# The 2022 M$_W$ 6.0 Gölyaka-Düzce earthquake: an example of a medium size earthquake in a fault zone early in its seismic cycle

P. Martínez-Garzón[1], D. Becker[1], J. Jara[1], X., Chen[1], G. Kwiatek[1] and M. Bohnhoff[1,2]

[1] Helmholtz Centre Potsdam GFZ German Research Centre for Geosciences, Potsdam, Germany.

[2] Free University Berlin, Institute of Geological Sciences, Berlin, Germany

Corresponding author: Patricia Martínez-Garzón (patricia@gfz-potsdam.de)

## Abstract

On November 23$^{rd}$ 2022, a M$_W$ 6.0 earthquake occurred in direct vicinity of the $M_W$ 7.1 Düzce earthquake that ruptured a portion of the North Anatolian Fault in 1999. The $M_W$ 6.0 event was attributed to a small portion of the Karadere Fault off the main North Anatolian Fault that did not rupture during the 1999 sequence. We analyze the spatio-temporal evolution of the $M_W$ 6.0 Gölyaka-Düzce seismic sequence at various scales and resolve the source properties of the mainshock. Modeling the decade-long evolution of background seismicity of the Karadere Fault employing an Epistemic Type Aftershock Sequence model shows that this fault was almost seismically inactive before 1999, while a progressive increase in seismic activity is observed from 2000 onwards. A newly generated high-resolution seismicity catalog from 1 month before the mainshock until six days after created using Artificial Intelligence-aided techniques shows only few events occurring within the rupture area within the previous month, no spatio-temporal localization process and a lack of immediate foreshocks preceding the rupture. The aftershock hypocenter distribution suggests the activation of both the Karadere fault which ruptured in this earthquake as well as the Düzce fault that ruptured in 1999. First results on source parameters and the duration of the first P-wave pulse from the mainshock suggest that the mainshock propagated eastwards in agreement with predictions from a bimaterial interface model. The $M_W$ 6.0 Gölyaka-Düzce event represents a good example of an earthquake rupture with damaging potential within a fault zone that is in a relatively early stage of the seismic cycle.

## Key Points

- Increased seismicity observed for Karadere fault after 1999 M$_W$ 7.4 Izmit earthquake while almost seismically inactive before.
- Aftershock distribution suggests activation of both the Karadere fault rupturing in this earthquake and Düzce fault that ruptured in 1999.
- Characteristics of P-wave first pulses suggest eastward rupture propagation in agreement with predictions from bimaterial interface models.

## 1. Introduction

Large strike-slip fault zones such as the San Andreas Fault in California, USA, or the North Anatolian Fault in Türkiye, among others, host some of the largest shallow earthquakes (typically up to M ~ 8, see e.g. Wesnousky et al., 1988; Stirling et al., 1996; Martínez-Garzón et al., 2015) worldwide. Some of these hazardous faults run near urban areas, and hence they have an associated risk. This is the case of the western portion of the North Anatolian Fault, being a seismic threat for the Istanbul metropolitan area and nearby population centers. There, the return period of M > 7 earthquakes rupturing the main fault trace has been estimated to be approximately 250 years

(Murru et al, 2016). The most recent large earthquakes along the NAFZ were the August 17th 1999 MW 7.4 Izmit and the November 11th,1999 MW 7.1 Düzce events (Fig. 1). Together they ruptured approximately 180 km and connected the Marmara segment in the west to the 1944 rupture in the east (Bürgmann et al., 2002; Sengör et al., 2005; Bohnhoff et al., 2013). On Nov 23rd 2022, a $M_W$ 6.0 earthquake occurred around 6 and 10 km away from the cities of Gölyaka and Düzce, respectively, and about 200 kilometers eastward of the Istanbul metropolitan area. In the following, we refer to this event as the Gölyaka-Düzce earthquake, felt especially in the province of Düzce and its districts (Eyidogan, 2022). Hypocenter locations provided by KOERI[1] and AFAD[2] reported that the Gölyaka-Düzce earthquake occurred close to the intersection of the ruptures of the 1999 MW 7.4 Izmit and MW 7.1 Düzce earthquake ruptures (Bouin et al., 2004; Konca et al., 2010, Fig. 1). Such M ~ 6 earthquakes are relatively infrequent in the region (according to the ISC-GEM[3] catalog, similar size events could have occurred in 1926 and 1944) but they hold potential to damage key infrastructure and lead to casualties. Hence, analyzing this earthquake and its pre- and post- seismic deformation is important to illuminate the local fault architecture (e.g. Ross et al., 2020), and recover how moderate events nucleate in the region (e.g. Malin et al., 2018; Durand et al., 2020). In addition, analysis of the earthquake source properties can help to decipher any preferential directions of seismic energy release, which is essential for seismic hazard estimation.

The Gölyaka-Düzce earthquake was the first M ~ 6 in the area after the 1999 M > 7 earthquakes and their aftershock sequences. As such, it represents the occurrence of an earthquake with damaging potential in a fault zone that is broadly still in the early stage of the seismic cycle, when the accumulated elastic strain is relatively low. However, at this location, the geometry of the fault zone is highly complex, with slip partitioning along two main fault traces bounding the Almacık Block and numerous secondary faults obliquely oriented (Fig 1c). Moment tensor estimation by different agencies (Table S1) consistently reported on a strike-slip mechanism with a small normal faulting component for this earthquake. The hypocenter location and fault plane appeared consistent with a portion of the Karadere Fault (Fig 1c). Different lines of evidence suggest that a at least a few km of the Karadere Fault did not rupture during the 1999 Izmit-Düzce sequence and thus remained loaded and then ruptured in the Gölyaka-Düzce earthquake (Bohnhoff et al., 2016b, Özalp and Kürcer, 2022). These include (i) the magnitude of the latest mainshock, (ii) the spatial extension of aftershocks and (iii) the previous surface mapping of local faults. First field surveys immediately after the $M_W$ 6.0 Gölkaya-Düzce earthquake found no surface rupture indicating that the slip did not extend to the surface (Özalp and Kürcer, 2022).

In this study, we analyze the spatio-temporal evolution of the Gölyaka-Düzce seismic sequence and the preceding seismicity in the area in detail with a focus on the source mechanisms and in context of the local seismotectonic setting. Our primary goals are to determine how the earthquake initiated, what are the ongoing deformation mechanisms in the region, and how the energy from this mainshock was radiated. Insights from this earthquake sequence are important to learn about the dynamics of potentially damaging earthquakes in complex transform fault settings, and the

---

[1] Kandili Observatory and Earthquake Research Institute. http://www.koeri.boun.edu.tr/new/en

[2] Disaster and Emergency Management Presidency  https://www.afad.gov.tr/

[3] ISC-GEM: International Seismological Centre    http://www.isc.ac.uk/iscgem/

rupture of highly stressed faults in immediate vicinity to a major strike-slip transform with
relatively low stress conditions corresponding to the fault zone being early in its seismic cycle.

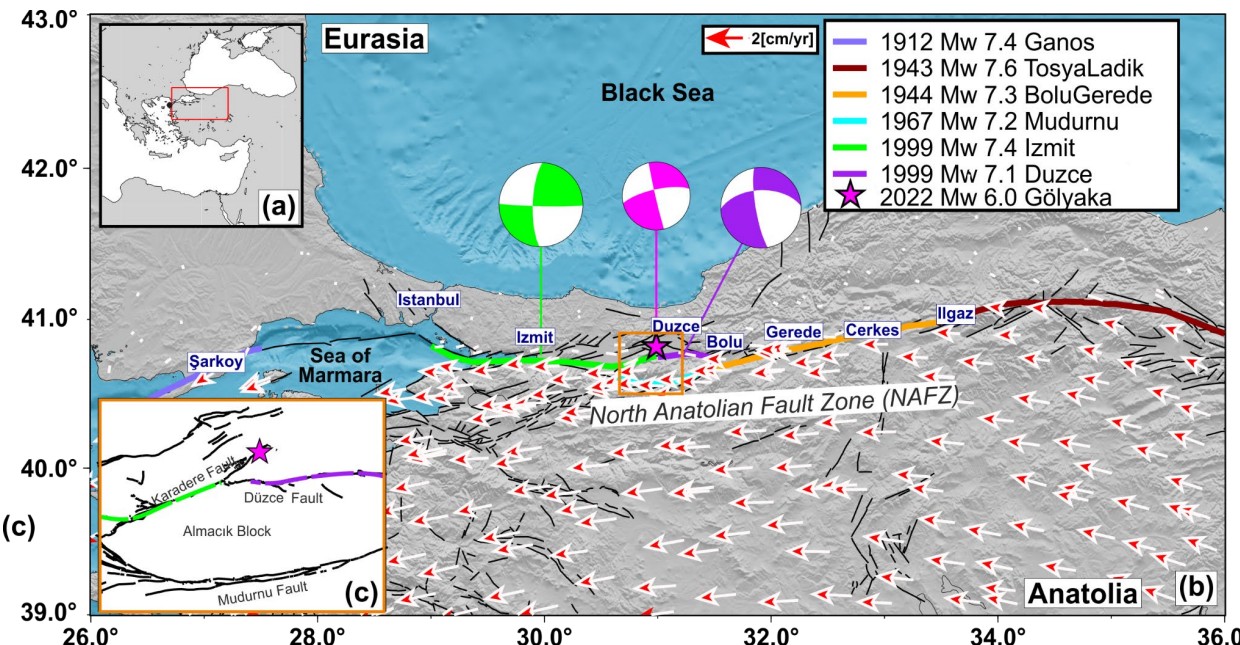

*Figure 1: Seismotectonic setting. (a) Location map with red rectangle indicating the area*
*enlarged in (b). The small black circle represents the location of the 2014 M 6.9 Saros earthquake.*
*(b) Study area with the location of main population centers along the North Anatolian fault zone*
*(NAFZ). Colored lines denote rupture extents of historical earthquakes along NAFZ, with their*
*respective magnitudes and dates indicated in the legend. Arrows are an updated GPS velocity field*
*for Türkiye (Kurt et al., 2022), considering the Eurasia-fixed reference frame. The magenta star*
*indicates the Gölyaka earthquake epicenter along with its focal mechanism from KOERI, as well*
*as the focal mechanisms of the Izmit (green) and Düzce (purple) earthquakes in 1999 (data from*
*the global CMT catalog, Ekström et al., 2012; Dziewonski et al., 1981. (c) into the region struck*
*by Gölyaka earthquake.*

## 2. Data and Methodology applied

### 2.1 Background seismicity evolution

To put the Gölkaya-Düzce seismic sequence in a regional and long-term context, we established a
consistent regional seismicity catalog between 1990 and 2022 for the two national seismic
agencies. The AFAD catalog has 31,081 events with magnitudes M [1.3 - 7.6] for such a period,
whereas the KOERI catalog reported 42,050 events M [1.4 - 7.4]. We analyzed the decade-long
evolution of the AFAD and KOERI regional seismicity catalogs through a declustering process
(Fig. 2). Both catalogs changed the reported magnitude type at the end of 2011 and beginning of
2012, from duration magnitude $M_d$ to local $M_L$. For consistency we homogenized both catalogs
converting uniformly to moment magnitude $M_W$, following the empirical relationships proposed
for the study region by Kadirioglu & Kartal (2016):

$$M_W = 0.7949M_d + 1.3420, \tag{1}$$

$$M_W = 0.8095 M_L + 1.303. \tag{2}$$

After both catalogs were homogenized to $M_W$, we estimated the magnitude of completeness $M^C$ of each catalog, following a probabilistic approach to fit the frequency-magnitude curve (Ogata and Katsura, 1993; Daniel et al., 2008, Jara et al., 2017). In contrast to the maximum curvature method, this technique fits a function to explain the magnitude-frequency distribution, including all catalog events. The number of earthquakes is fit as a function of magnitude as follows:

$$N(m) = A \times 10^{-bm} \times q(m), \tag{3}$$

where the $b$-value represents the slope of the Gutenberg-Richter law, $A$ is a normalization constant, and $q(m)$ is the probability that one earthquake of magnitude $m$ is listed in the catalog . Then, we modeled $q$ as (Ogata and Katsura, 1993):

$$q(m) = \frac{1}{2} + \frac{1}{2} erf\left(\frac{m - \hat{\mu}}{\sqrt{2}\hat{\sigma}}\right), \tag{4}$$

where $erf$ is the error function and $\hat{\mu}$ and $\hat{\sigma}$ correspond to the mean and standard deviation of the probability distribution function, respectively. We optimized $[A, b, \hat{\mu}, \hat{\sigma}]$ for each catalog following a Bayesian approach to derive the parameters posterior Probability Density Function (PDF). We used the Markov Chain Monte Carlo sampler of PyMC (Salvatier et al., 2016), to draw 500.000 samples from the posterior PDF. The inferred parameters and their associated uncertainties are in Table S2. Then, the completeness magnitude $M^C$ is computed as follows:

$$M^C = \hat{\mu} + 2\hat{\sigma}, \tag{5}$$

i.e., a 97.7% probability threshold, which yielded a $M^C = 3.4$ for the AFAD catalog, whereas, for the KOERI one, we obtained a $M^C = 4.1$ (see insets in Fig 2, and Figs S1 and S2 for the obtained fitting). Once $M^C$ was estimated for both catalogs, we declustered them using an epidemic-type aftershock sequence model (Marsan et al., 2017; Jara et al., 2017). Such approach considers the total seismicity rates $\lambda(x, y, t)$ as the following sum:

$$\lambda(x, y, t) = \mu(x, y, t) + \nu(x, y, t), \tag{6}$$

where $\nu(x, y, t)$ accounts for the aftershock productivity, and $\mu(x, y, t)$ is the background seismicity rate for earthquakes occurring at a given location $(x, y)$ and time $t$. The aftershock rate was estimated following the Omori-Utsu law pondered by a power spatial density, following:

$$\nu_i(x, y, t) = \sum_{i \vee t_i < t} \frac{\kappa(m_i)}{(t + c - t_i)^p} \frac{(\gamma - 1)L(m_i)^{\gamma - 1}}{2\pi\left((x - x_i)^2 + (y - y_i)^2 + L_i^2\right)^{\frac{\gamma + 1}{2}}}, \tag{7}$$

in which $c$, $\gamma$, and $p$ are constants, and $\kappa(m)$ is the productivity law with a constant $\alpha$ [Ogata, 1988]. $L(m) = L_0 \times 10^{0.5(m - M^C)}$ is the characteristic length in km (Utsu & Seki, 1955; van der Elst & Shaw, 2015). Here, we imposed realistic values for parameters $\alpha = 2$, $p = 1$, $c = 10^{-3}$ days, $\gamma = 2$, and $L_0 = 1.78$ km (Marsan et al., 2017; Jara et al., 2017; Karabulut et al., 2022). Parameters $\kappa$ and $\mu(x, y, t)$ were inverted. The background seismicity was computed as follows:

$$\mu(x,y,t) = \sum_i \frac{\mu(x_i,y_i,t_i)}{\lambda(x_i,y_i,t_i)} e^{\frac{-\sqrt{(x-x_i)^2+(y-y_i)^2}}{l}} e^{-t-t_i \vee \tau} \times \frac{1}{2\pi l^2 a_i}, \qquad (8)$$

with $l$ and $\tau$ space and time being smoothing parameters, and $a_i = 2\tau - \tau\left(e^{\frac{t_s-t_i}{\tau}} - e^{\frac{t_e-t_i}{\tau}}\right)$, where $t_s$
and $t_e$ are the temporal beginning and end of the catalog, respectively. $\kappa$ is inferred as:
$$\kappa = \frac{\sum_i \quad 1 - \frac{\mu(x_i,y_i,t_i)}{\lambda(x_i,y_i,t_i)}}{\sum_i \quad e^{\alpha m_i \left(ln(t_e+c-t_i)-ln(c)\right)}}. \qquad (9)$$

Here, we used a smoothing length $l$ =100 km and a smoothing duration $\tau$ = 100 days. Such choices
are able to preserve the potential accelerations/decelerations from catalogs (Marsan et al., 2017;
Jara et al., 2017). We declustered the catalogs using the obtained $M^C$ for each catalog and the fault
regions in Fig 2. When doing so, we observed an apparent increase in the seismicity from the
AFAD declustered catalog around 2012 (Fig S1). Around that time, AFAD changed the reported
magnitudes from $M_d$ to $M_L$. Although we converted the corresponding magnitudes to $M_W$, this
change in the magnitude estimation might still produce spurious acceleration/deceleration in the
background seismicity rate. We then tested higher $M^C$ values, finding that such behavior
disappears around $M^C = 4.1$ (Fig S1). Thus, we finally used $M^C = 4.1$ for both catalogs (Fig 3a,
b). The final parameters utilized for each catalog are provided in Table S1.

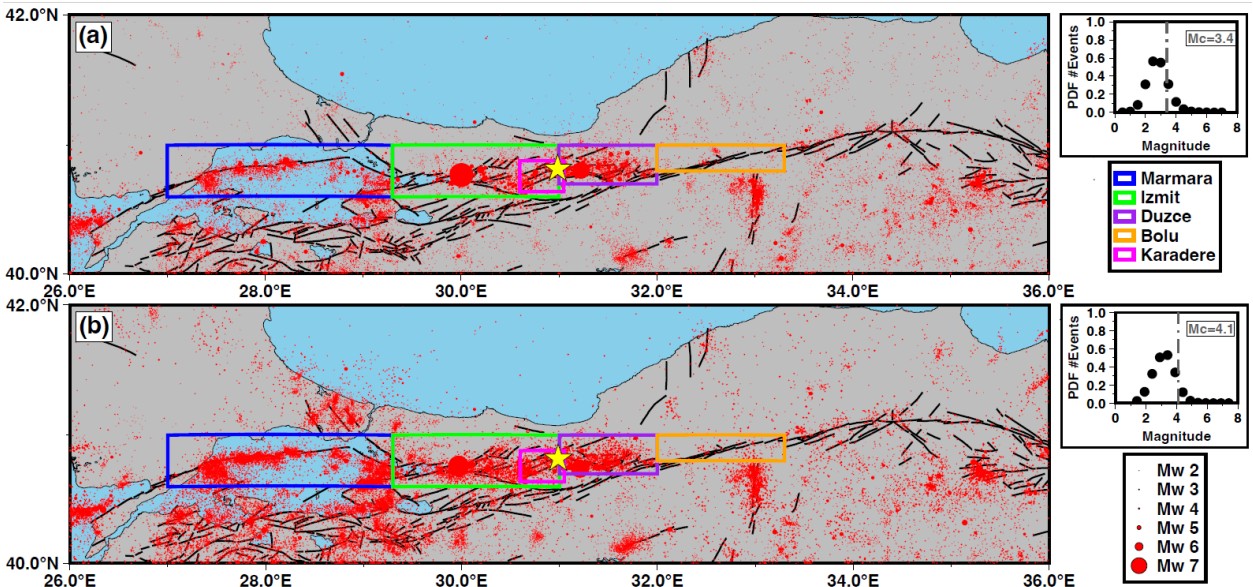


**Figure 2:** *Regional Seismicity. (a) AFAD catalog from 1990 to 30/11/2022. Yellow star denotes*
*the $M_W$ 6.0 Gölyaka-Düzce epicenter. Color boxes indicate the target regions where the seismicity*
*is analyzed. Right:Catalog's Probability Distribution Function (PDF), where the vertical dashed*
*line denotes the $M^C$. (b) Same as (a), but for KOERI catalog. See Figs. S3 and S4 for the spatial*
*distribution of seismicity inside each region.*
## 2.2 Enhanced seismicity catalog framing the mainshock
To generate an optimized enhanced seismicity catalog with lowest possible magnitude detection
threshold around the $M_W$ 6.0 Gölkaya-Düzce earthquake epicenter, we processed continuous
waveform recordings from 16 local seismic stations and 9 local accelerometer stations. We
covered a time period from one month before the mainshock up to almost 6 days after it (October
23$^{rd}$, 2022 at 00:00h up to November 29$^{th}$, 2022 at 00:00h. The employed stations belong to the
AFAD and KOERI seismometer and strong motion networks.
We detected P- and S- wave onset times embedded in the continuous recordings applying the
supervised Artificial Intelligence method Phasenet (Zhu and Beroza, 2019) trained on the
seismicity database from Northern California. This method has proven to improve the detection
process especially of small earthquakes (e.g. Martinez-Garzon et al., 2023). With this method,
148,948 body wave onsets were detected, out of which 78,410 were detections of P-waves and
70,568 were detections of S-waves.
The P- and S- picks were associated with seismic events using the unsupervised technique
GAMMA (Zhu et al., 2022). To classify an event to be an earthquake, a minimum of 4 necessary
picks (either P and/or S) was set. The picks were spatio-temporally clustered using the Density
Based Spatial Clustering of Applications with Noise (DBSCAN) method. About 19% of the total
amount of picks were associated with earthquakes. This way, we have obtained a catalog of
detections containing 3,361 possible seismic events (Fig 3a).
As a comparison, KOERI reported a total of 505 events with magnitudes $\geq M_L$ 0.5 for the same
spatio-temporal region analyzed here (Fig 3b). Out of them, 440 correspond to common events
from both catalogs.
In the next step, the waveforms from all events corresponding to the period before the $M_W$ 6.0
mainshock were visually inspected. About 343 detections from the time period before the
mainshock were removed as they showed signals in only one or two of the accelerometers,
typically exhibiting $t_S - t_P > 5\ s$, which is larger than what is expected for a small local event.
Additional 35 events were identified as regional events with locations outside the study region,
and additional 11 events were identified as duplicates and removed. In the following, we refer to
this catalog as the "*catalog of detections*".
We calculated event locations by employing the probabilistic location software NLLoc (Lomax et
al., 2000; 2009). Here, only events with a minimum of 6 P- and/or S- picks were further processed,
which implicitly removes possible false signal associations with less than 6 phases from the catalog
of detections. The local 1-D velocity model from Bulut et al. (2007) was employed assuming a
constant $v_p/v_s$ ratio of 1.73. The search area encompassed a 400 km x 200 km region centered
around the mainshock epicenter. In the following, we refer to this refined catalog as the "*catalog
of absolute locations*". Further details on the refining of the catalog of absolute locations are
provided in Text S1. This way, we obtained a catalog of 1,290 events with absolute locations,
containing 8,927 P-wave picks and 7,822 S-wave picks for further processing (Fig. S5). In this
catalog, the median errors in the x-, y- and z- directions are 2.3 km, 3.1 km and 3.4 km,
respectively.
In the next step, a relative event relocation was performed using hypoDD (Waldhauser &
Ellsworth, 2000; Waldhauser et al., 2004). We utilized both catalog differential travel times
derived from the automatic Phasenet P- and S-picks and cross-correlation time differences derived
from the event waveforms. To estimate the waveform cross-correlations, we employed time
windows covering 1 s and 2 s centered at the P- and S-onset, respectively. The waveforms were
filtered with a 3ʳᵈ order Butterworth bandpass filter between 2 and 10 Hz. The retrieved correlations
with a normalized cross-correlation coefficient of at least 0.7 were kept and the square of the cross
correlation coefficient was used as weight in the relocation procedure. To look at the spatio-
temporal evolution of the seismicity, we demanded a minimum of 8 catalog time differences (either
P- and/or S-phases) for each event combination resulting in a catalog of 918 relocated events. In
the following, we refer to this further refined subset as the "*relocated catalog*". The median formal
relative relocation errors in the x-, y-, and z- directions are 11 m, 13 m, and 12 m, respectively.

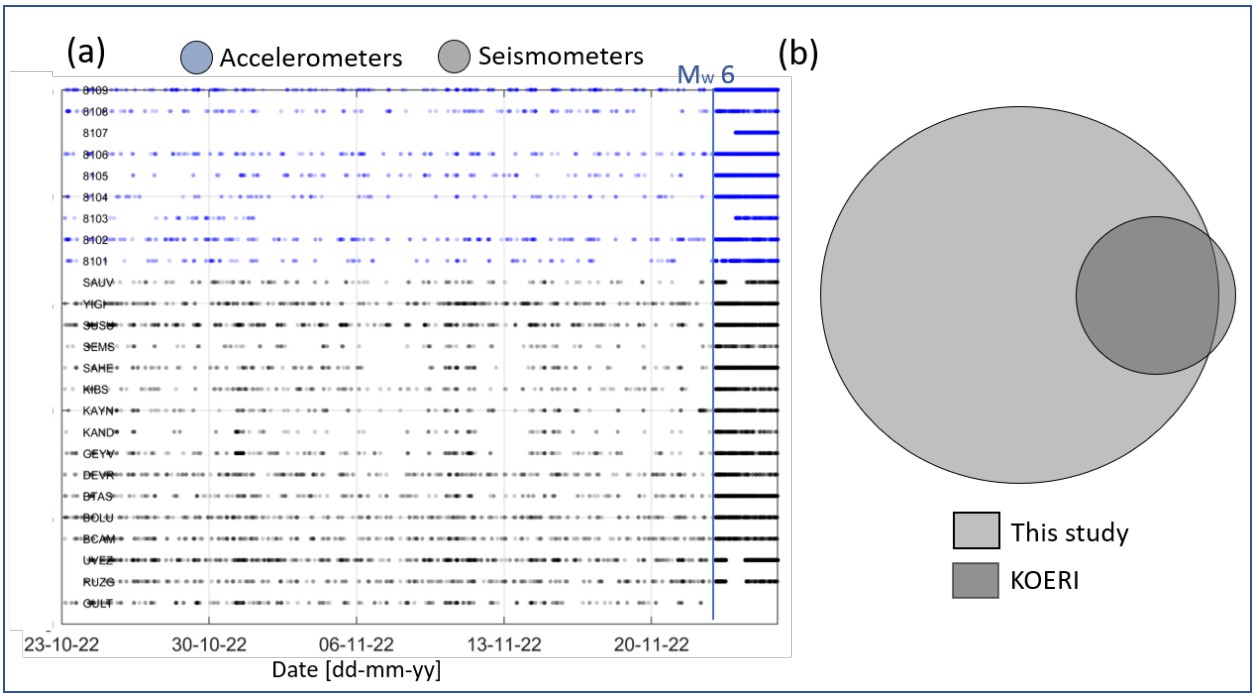


**Figure 3: Picks and detections from the AI-aided catalog**. *(a) Associated picks as a function of time, per station. Vertical blue line marks the $M_W$ 6.0 Gölyaka-Düzce earthquake. (b) Venn diagram showing the earthquakes included in our catalog of detections vs the events included in the KOERI catalog for the same spatio-temporal region.*

## 2.3 Earthquake source parameters and directivity

The estimation of point source parameters of the $M_W$ 6.0 mainshock was performed using the
spectral fitting method (Kwiatek et al., 2015). We used 249 high-gain seismometers of Kandilli
Observatory (KO) and the National Seismic Network of Türkiye (TU). The combination of these
networks offers sufficient azimuthal coverage and signal-to-noise ratio for the analyzed catalog.
From these networks, stations with epicentral distances of 200 - 800 km are used to derive source
parameters of the mainshock.
Three-component ground velocity waveforms from stations with a signal-to-noise ratio of at least
4 dB were filtered using a 2ⁿᵈ order 0.02 Hz high-pass Butterworth filter and then integrated in the
time domain. We utilized window lengths of 25 s from the P- and S- wave ground displacements,
allowing for additional 4 s before the P- or S- wave onsets. The selected window length of 25s in
combination with the minimum station distance of 200 km prevents contamination of the P-wave
window by S-wave energy as the S-P time difference is larger than 25 seconds assuming average
velocities and vp/vs-ratios along the travel path. The edges of the selected windows were smoothed
using von Hann's taper. We utilized the three components from the far-field ground displacement
spectra using the Fourier transform and then combined altogether. The observed ground
displacement spectra were fit to Brune's point-source model expressed as:

$$u^{th}(f) = \frac{\langle R_C \rangle}{4\pi\rho V_C{}^3 R} \frac{M_0}{1 + \left(\frac{f}{f_c}\right)^2} exp\left(\frac{-\pi f R}{V_c Q_c}\right), \qquad (10)$$

where $R$ is the source-receiver distance, $M_0$ is the seismic moment, $f_c$ is the corner frequency
where $c$ represents either the P- or S- wave, $Q_c$ is the quality factor, and $< R_c >$ is the average
radiation pattern correction coefficient of either P- or S- waves. Following Boore and Boatwright
(1984), we applied $R_P$=0.65 and $R_S$=0.7 for P and S waves, respectively, that are representative
constants for the regional strike-slip faults. We used $V_P$ = 5680 m/s and $V_S$ = 3280 m/s (from Bulut
et al., 2007, averaged at the depth interval where the earthquakes occurred), assuming $V_P/V_S$ = 1.73
and a density $\rho$ = 2700 kg/m³. We inverted for [$M_0$, $f_c$, $Q_c$] optimizing the cost function:

$$\|log_{10}u^{obs}(f) - log_{10}u^{th}(f)\|_{L1} = min, \qquad (11)$$

where $u^{th}(f)$ and $u^{obs}$ are the theoretical and observed ground displacement amplitude spectrum
for a given station and P or S wave. The starting model for $M_0$ and $f_c$ was taken using Snoke's
integrals (Snoke, 1987) and we assumed initial values of Q = 400 for both P- and S- wave trains.
We utilized a grid search optimization (assuming starting model) followed by simplex algorithms
(starting from the best   model of a grid search). Source parameters that deviated from the average
by more than three standard deviations were eliminated from the calculation. The final source
parameters (i.e. seismic moment, corner frequency, quality factor) were calculated as average
values from all stations.
In the following, we calculated the static stress drop using the formula valid for a rectangular
strike-slip fault (Shearer 2009):

$$\Delta\sigma = \frac{2}{\pi} \frac{M_0}{W^2 L}, \qquad (12)$$

where $W$=$8\ km$ represents the fault width, assumed from the depth extent of the aftershocks (see
Section 3.2.2). The rupture length $L$ is estimated as double the    source radius while    using
Brune's source model constants and, for comparison, the Haskell's rectangular source assuming
$V_R$=0.9$V_s$ (see Savage et al., 1972, Table 1 for details) using the equation

$$L = \frac{C_C V_C}{2\pi f_c} \qquad (13)$$

in which $C_C$ is the geometrical correction coefficient ($C_P$= $C_S$ = 4.7 for Brune's model, and $C_P$ =
1.2, $C_S$ = 3.6 for the Haskell's model, see Savage et al., 1972) and $f_C$ and $V_C$ represent    the
corner frequency and seismic velocity of either P- or S-waves, respectively.
For the earthquakes comprising the absolute locations catalog, we estimated only moment
magnitudes $M_W$ using a simplified approach following    Snokes's (1987) integrals:

$$J_S = 2\int [u^{\cdot}(f)]^2 df, \qquad (14)$$

$$K_S = 2\int [u(f)]^2 df, \qquad (15)$$

where $u^{\cdot}(f)$ and $u(f)$ are ground velocity and displacement S-wave spectra corrected for
attenuation and prepared from S-wave waveforms processed in the same way as for the mainshock.
The original seismograms were filtered with 1 Hz high-pass 2nd order Butterworth filter and we

used a shorter 5 s window framing the first S-wave arrival to limit the influence of low-frequency noise for predominantly small earthquakes (M<4). The integrals in eq. (14) and (15) were corrected for the finite frequency band following di Bona and Rovelli (1991). The seismic moment has been estimated with (Snoke, 1987):

$$M_0 = 8\pi\rho V_S{}^3 R \left(\frac{K_S{}^3}{J_S}\right)^{0.25}, \tag{16}$$

and the moment magnitude was calculated using the standard relation (Hanks and Kanamori, 1976):

$$M_W = \frac{(log_{10} M_0 - 9.1)}{1.5}. \tag{17}$$

Similar to the mainshock, for each event the final seismic moment and moment magnitude were calculated as average values from all stations containing S-wave arrivals. Due to the limited number of S-waves with sufficient signal-to-noise ratio for the smallest earthquakes, the uncertainties were estimated using the mean absolute deviation.

Large earthquake ruptures potentially involve a propagation process along a fault plane. The rupture propagation direction could be deduced from the azimuthal variations of amplitude and frequency content of the apparent source time functions (ASTF) (Stein and Wysession, 2003) providing important information for seismic risk assessment. For a unilateral rupture, ideally this would lead to shorter ASTFs displaying larger amplitudes in the direction of rupture propagation, and longer duration and smaller amplitude ASTFs in the opposite direction. To obtain the ASTFs, we initially tested the application of Empirical Green's Function (EGF) technique and tried four EGF candidates (Table S3) to recover the directivity of the mainshock (see text S2 for all details) that led to inconclusive results.

We therefore tested the azimuthal variations in the duration and frequency content of the initial P-wave arrivals for seismometers located at epicentral distances of 50-100 km from the mainshock. For comparison, we additionally included integrated signals from accelerometers located at much closer distances. We only used unclipped first P-wave pulses that were rotated into the radial direction from 3-component seismograms to enhance the signal-to-noise ratio of the initial portions of the P-wave. The first P-wave pulses contain a combination of information including the source time function and effects related to wave propagation. However, comparing P-wave pulse characteristics for stations located at similar distances from the mainshock epicenter allows us to suppress propagation effects. Therefore, the initial portion of the seismogram can be taken as a proxy for the ASTFs. Variation in rise time and duration of the P pulses can then be used to infer whether the earthquake displays rupture directivity.

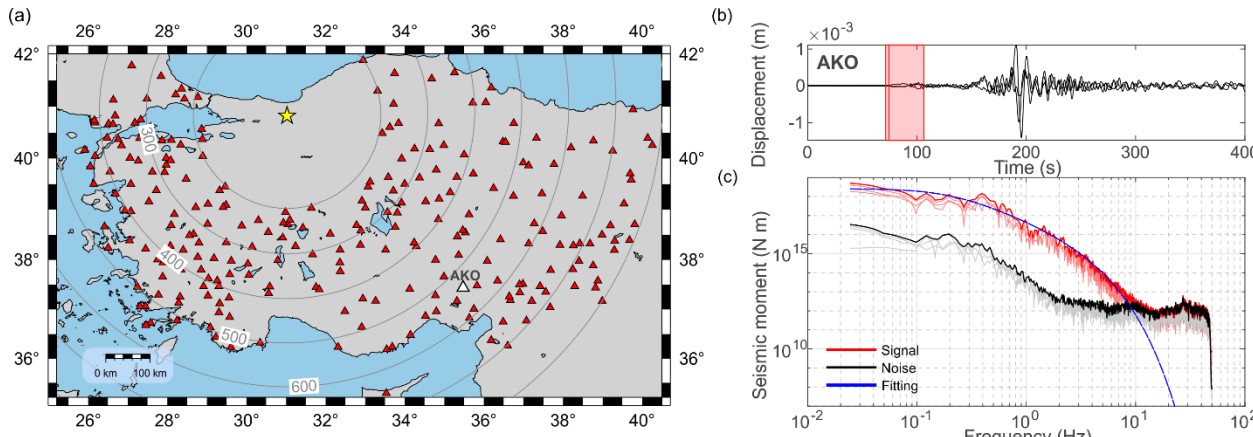

303

**Figure 4**. **Source parameter analysis**. *(a) Station distribution employed for the source parameter estimation (red upward triangles). These stations lie within a source-station distance between 200 km and 800 km. Yellow star shows the 2022 $M_W$ 6.0 Gölyaka-Düzce mainshock. (b) Three-component displacement waveforms for the mainshock recorded by station AKO (with epicentral distance ~530 km, see white triangle in (a)). Red rectangle highlights the employed P-wave window. (c) Displacement spectrum of the P-wave signal (red) and the noise before the signal (black). Blue thick line indicates the modeled spectrum yielding the source parameters: $M_0 = 2.62 \times 10^{18}, f_c = 0.19 Hz, Q_P = 308$.*

## 3  Results

### 3.1 Long-term evolution of background seismicity in the Gölyaka-Duzce region

We analyzed the evolution of the background seismicity along defined segments of the NAFZ including the Marmara, Izmit, Düzce, Bolu, and Karadere segments (Fig. 2). Both national Turkish catalogs introduced above show that the Bolu segment displays a low background seismicity rate when compared to e.g. Izmit or Marmara segments (Fig 5). Aseismic slip (surface creep) has been reported to occur along this segment, occurring for at least 70 years (Ambraseis, 1970; Cakir et al., 2005; Cetin et al., 2014; Bilham et al., 2016, Jolivet et al., 2023). This might be a possible explanation for the low seismicity rate. The Marmara, Izmit, and Düzce segments appear to host a constant background seismicity rate with time, especially after the 1999 Izmit and Düzce sequence (Fig 5). Both catalogs report a deceleration of background seismicity after the 2014 $M_W$ 6.9 Saros earthquake (Bulut et al., 2018), supporting the idea that some significant deformation process not yet understood in detail was affecting the seismicity along the NAFZ (Karabulut et al., 2022).

The Karadere fault hosted a comparatively low background seismicity before the 1999 Izmit and Düzce earthquake sequence. A change in its seismic behavior is observed afterwards, when this segment experienced an increase of the seismic activity including more than 5 events with M > Mc = 4.1. The shape of the background rates is different for the AFAD and KOERI catalogs. This difference might be due to the different number of seismic stations operated by the agencies in this area, hence affecting the monitoring capabilities and detection thresholds. Therefore, it is likely that the region was tectonically activated by the earthquake sequence in 1999, and progressively loaded since then, leading to the $M_W$ 6.0 Gölyaka-Düzce earthquake 23 years later. Interestingly,

the region did not exhibit a lower background rate after the 2014 $M_W$ 6.9 Saros earthquake, different to the other NAFZ segments in the area.

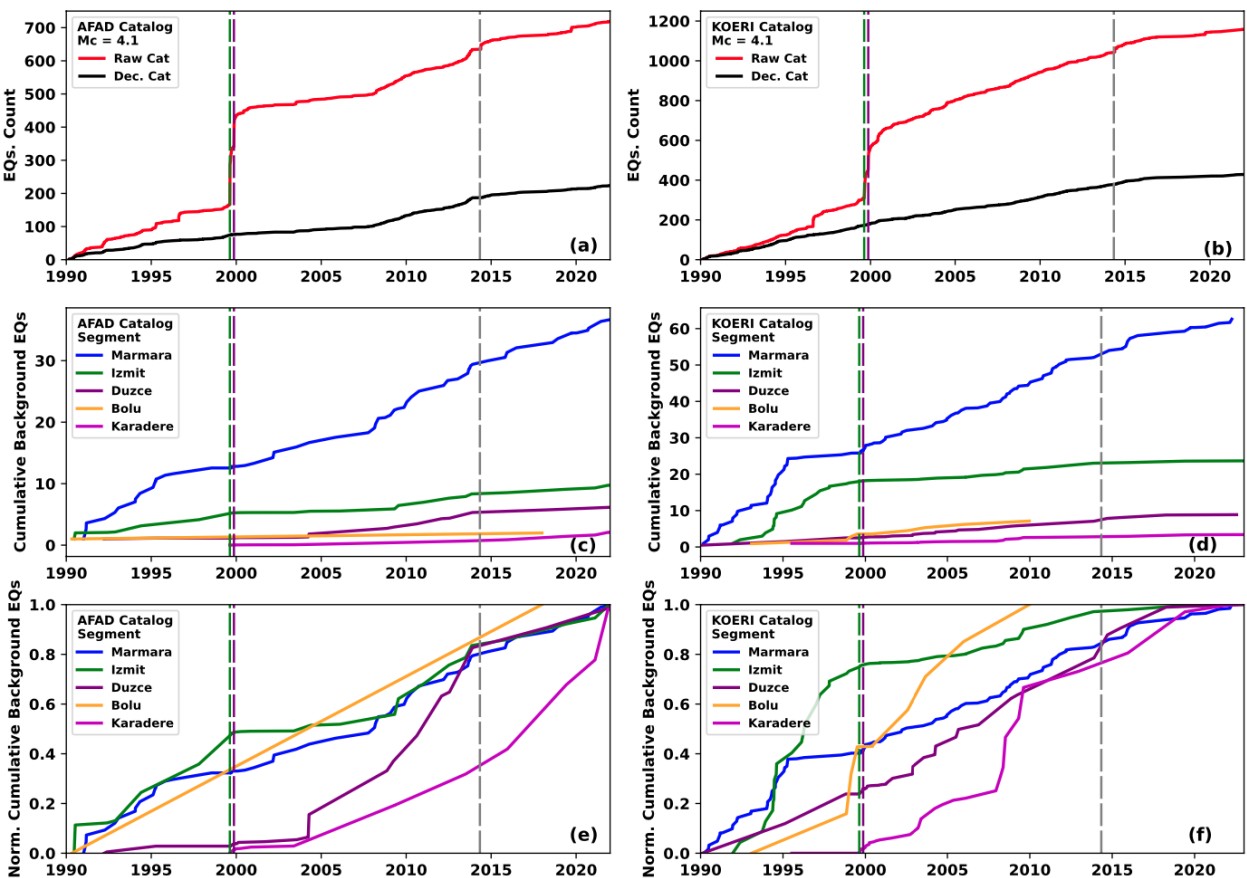

**Figure 5:** *Temporal evolution of the background seismicity at different segments of the NAFZ.*
*(a) Complete (black) and declustered (red) AFAD catalog using $M^C$=4.1. (b) Same as (a) but with*
*KOERI catalog. (c) Cumulative background seismicity, color-coded by region as in Fig. 2, for the*
*AFAD catalog. (d) same as (c) for the KOERI catalog. The vertical green, magenta and grey lines*
*represent the time of occurrence of the $M_W$ 7.4 Izmit, $M_W$ 7.1 Düzce, and $M_W$ 6.9 Saros*
*earthquakes, respectively. (e) Normalized cumulative background seismicity, color-coded by*
*region as in Fig. 2, for the AFAD catalog. (d) same as (c) for the KOERI catalog.*

## 3.2 Spatio-temporal seismicity distribution before and after the 2022 Gölyaka-Düzce earthquake

We obtained an enhanced seismicity catalog with 1,290 refined hypocenter locations as described in Section 2.2 covering the area of longitude [30-32°E] and latitude [40-42°N] for the time period [October 23$^{rd}$, 2022 at 00:00h up to November 29$^{th}$, 2022 at 00:00h] and including moment magnitudes as low as $M_W$ 0.7. Out of them, a total of 222 and 1,032 seismic events correspond to events preceding and following the 2022 Gölyaka -Düzce mainshock, respectively. For the same region and time interval, the seismicity catalog provided by the KOERI agency contained 529 events, out of which 23 and 506 corresponded to events preceding and following the mainshock, respectively (Fig 3b).

Using a goodness of fit method (Wiemer and Wyss, 2000), the magnitude of completeness of the derived catalog within selected region is $M_W^c$ = 1.5. Calculating the *b*-value for events above $M^C$,

from both the periods before and after the mainshock, we find a value of $b = 0.95 \pm 0.05$ (Fig. S6).
This could be related to the fact that we utilized $M_W$ while many other estimates use $M_L$ that may
lead to larger $b$-value (see e.g. Raub et al., 2017). Alternatively, the relatively low $b$-value may
suggest that the fault did not yet release all its accumulated strain (e.g. Gulia & Wiemer, 2019).
Given the magnitude of the mainshock and the spatial extent of the rupture we consider the latter
option as rather unlikely.

### 3.2.1 Seismic activity preceding the Gölyaka-Düzce earthquake

The $M_W$ 6.0 Gölyaka-Düzce earthquake hypocenter is located at the northeastern portion of the
Karadere fault that remained unbroken during the 1999 $M_W$ 7.4 Izmit and 1999 $M_W$ 7.1 Düzce
events.
The area that ruptured in the $M_W$ 6.0 Gölyaka-Düzce earthquake and its surroundings only
displayed a small number of seismic events during the 30 days preceding the mainshock. The
catalog of absolute locations reported 222 seismic events during this time, out of which 55 could
be successfully relocated. Most of the relocated seismic activity occurred away from the future
$M_W$ 6.0 earthquake rupture, extending up to 50 km to the East (Fig. 6). The locations of these
seismic events show a good correspondence with the mapped local faults (Emre et al., 2018). A
small cluster of events is visible at the eastern edge of the analyzed region, coinciding with the
termination of a local fault, near a quarry area (see Fig. 6a for location). The presence of a quarry
in the area suggests that some of these events could be quarry blasts. However, these events appear
to be regular seismic events based on the following: (i) these detections display regular P- and S-
wave trains, (ii) their hypocentral depth is deeper than 8 km, and (iii) these events occur randomly
in time. Within a 25 km radius from the epicenter of the $M_W$ 6.0 Gölyaka-Düzce earthquake, 23
events were included in the catalog of absolute locations. The most active time period was between
Nov 6th and 11th, where a small spatially clustered seismic sequence with magnitudes up to $M_W$
2.2 occurred about 7.5 km to the North of the mainshock epicenter (Fig 6, Fig S7). The location
of this cluster coincides with the deepest part of the fault activated with the aftershock sequences.
Both the catalog of detections and the catalog of absolute locations show that seismicity rates were
time-invariant with a transient increase in seismic activity around Nov 10th reflecting the transient
cluster North of the future mainshock. This increase in the seismicity rates quickly decayed back
to the level before the occurrence of the cluster, remaining constant until the occurrence of the
mainshock (Fig 7). The regional seismicity did not display any significant acceleration at the scale
of days to hours before the mainshock.

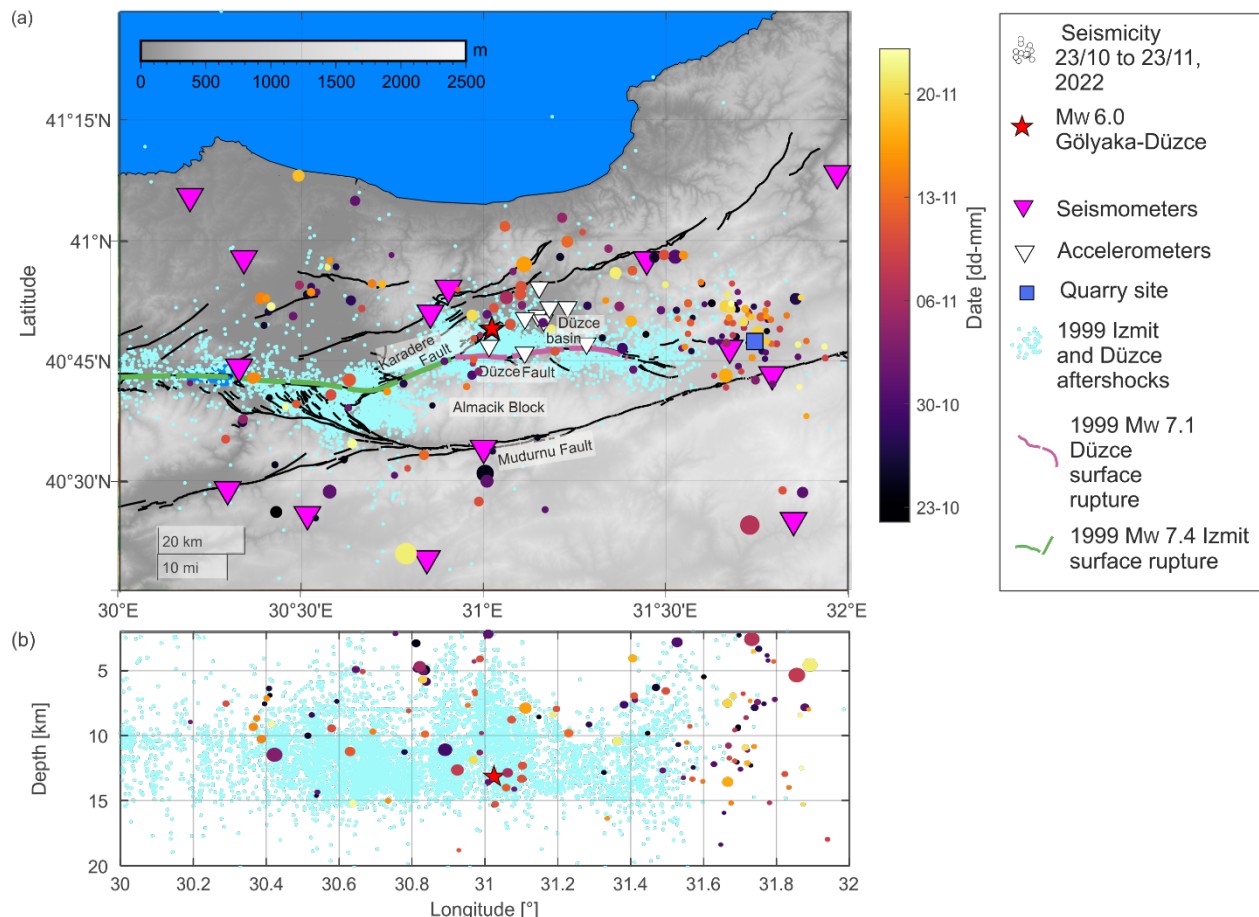


*Figure 6: Seismicity located during the preceding month. Seismicity distribution included in the absolute location catalog (colored circles) during the month preceding the Gölyaka-Düzce earthquake (red star). Symbol size is encoded with magnitude. Surface ruptures of the 1999 $M_W$ 7.4 Izmit and $M_W$ 7.1 Düzce earthquakes are shown with green and pink dashed lines, respectively. For comparison, seismic activity for three different time periods around the 1999 Izmit and Düzce mainshock is shown in cyan (from Bohnhoff et al, 2016b; Bulut et al., 2005). Fault traces are from Emre et al., (2018).*

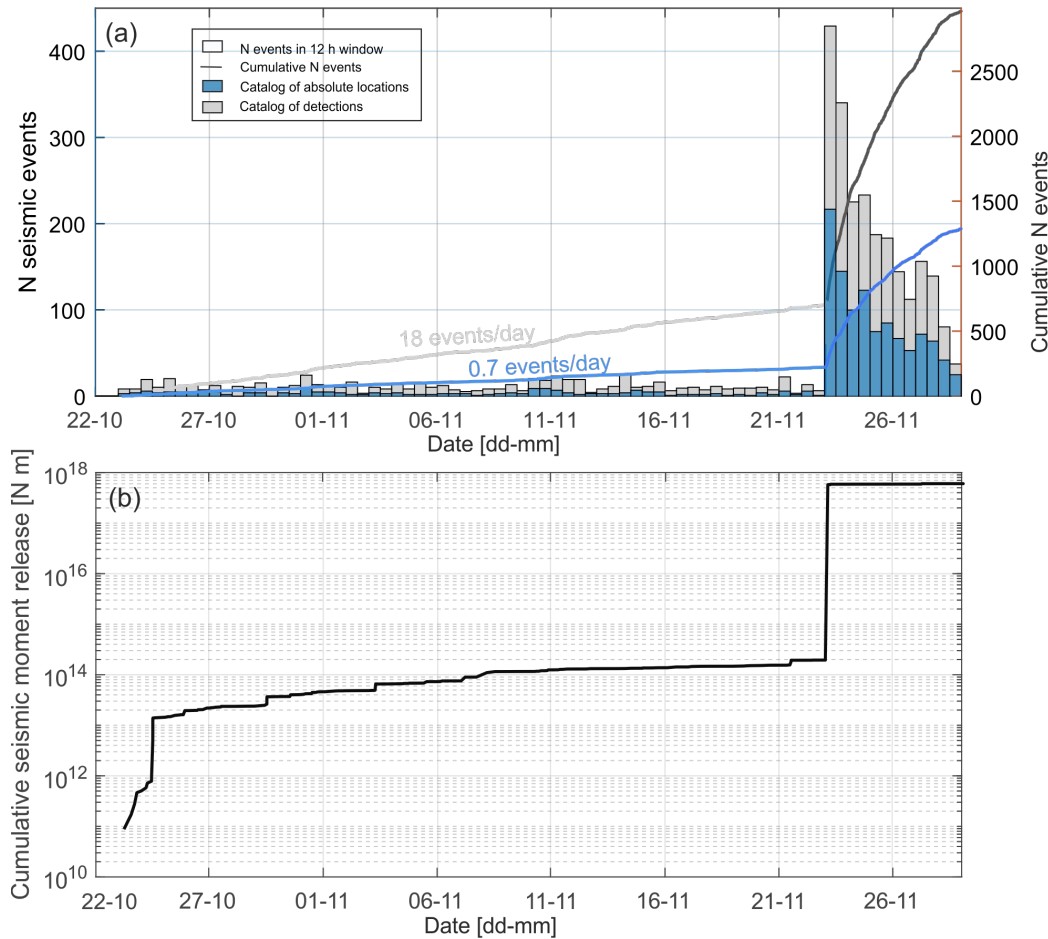

395

**Figure 7**: *Temporal evolution of seismic activity and seismic moment. (a) Bars: Histogram of seismicity rates, where every bar represents a time period of 12h. Grey and blue colors represent the seismicity included in the catalog of detections and absolute locations, respectively. Lines: Cumulative number of seismic events as a function of time. Lighter and darker colors represent the time periods before and after the mainshock (b) Evolution of cumulative seismic moment release from the catalog of absolute locations.*

### 3.2.2 The aftershock sequence following Gölyaka-Düzce earthquake

After the $M_W$ 6.0 Gölyaka-Düzce earthquake, vigorous seismic activity struck the region during the following days. Compared to the scattered seismicity in a much larger region, most of this early aftershock activity occurs within an area extending 15 km to the East and West as well as 8 km to the North and South of the mainshock epicenter, respectively (Figs. 8, 9). Generally, aftershocks typically occur around the mainshock rupture area, and they may also activate nearby-faults due to stress changes induced by the mainshock. In first order approximation the relocated aftershock activity delineates a planar structure trending SW-NE that is dipping towards the NNW, consistent with the geometry of the Karadere fault (Fig. 9). The plane best fitting to the seismicity (contained within 1 km distance) has a strike of $\varphi = 257°$ and a shallow dip of approximately $\delta = 45°$ (Fig. 9). The strike of this plane is thus in good agreement with the moment tensor solutions for the $M_W$ 6.0 mainshock (Table S2). However, the dip of our plane is shallower than the $\delta^{fm} = 72° - 82°$ reported by the moment tensor solutions (Fig. 9c). The depth of the seismicity along the strike of the fault segment is not uniform, with the southwestern portion of the fault displaying generally shallower seismicity from 5 to 13 km depth, and the northeastern portion of the activated

fault between 9 and 16 km depth (Fig 9b). Along strike, the hypocentral location of the mainshock coincides with this depth change, suggesting the presence of a fault jog or a heterogeneity that could have promoted a stress concentration.

The mainshock triggered an aftershock sequence that within the first six days can be fitted with an Omori law of the shape $N(t) = kt^{-p}$, with $p = 0.90$ and $k = 2.5$ (see Fig S8). Typical values observed for the $p$-value representing the decay rate oscillate around 1.0, suggesting that the aftershock decay associated with this sequence is fairly standard, including 3-4 $M_W > 4$ earthquakes occurring within the analyzed time window.

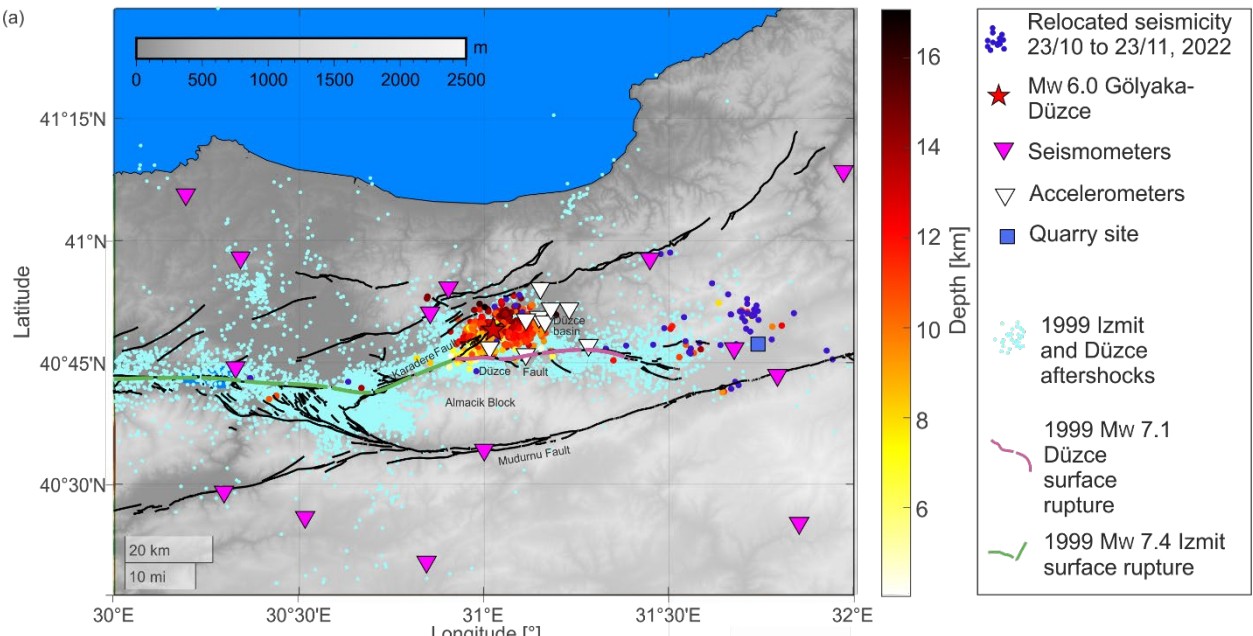

*Figure 8: Seismicity distribution after the Gölyaka-Düzce earthquake (colored dots). Cyan dots in the background reflect 1999 Izmit and Düzce aftershocks. For comparison, seismic activity for three different time periods around the 1999 Izmit and Düzce mainshock is shown (from Bohnhoff et al, 2016b, Bulut et al., 2005).*

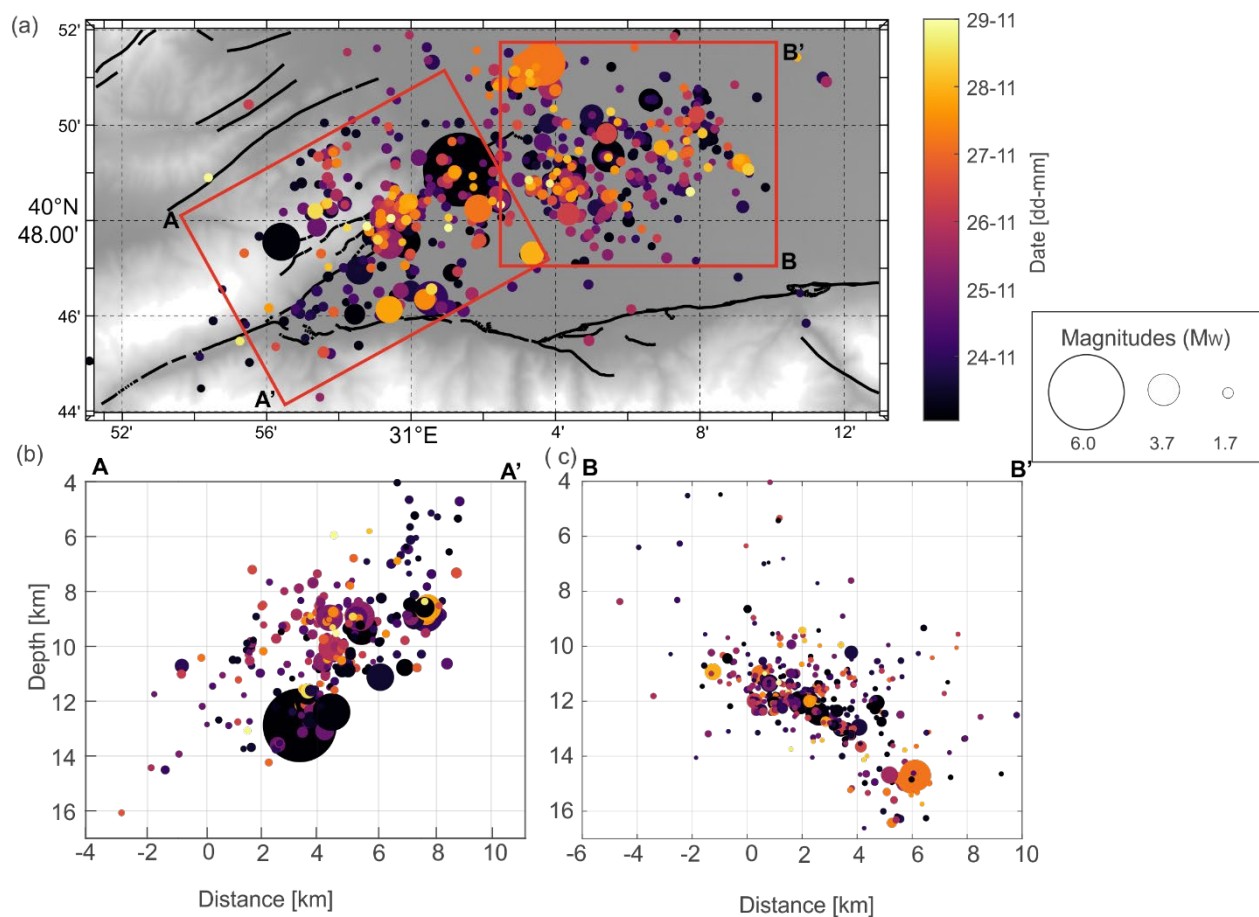

**Figure 9: Zoom on the spatio-temporal distribution of the seismicity during 6 days following the Gölyaka-Düzce earthquake.** *(a) Map view. Depth profiles along (b) A- A' (approximately perpendicular to the Karadere fault strike), and (c) B-B' (approximately perpendicular to the strike of the Düzce fault). Symbol size and color are encoded with magnitude and date, respectively.*

### 3.3 Source parameters and directivity of the Gölyaka-Düzce mainshock

Earthquake source parameters for the 2022 $M_W$ 6.0 Gölyaka-Düzce mainshock are provided in Table 1, with the average values and multiplicative error factors calculated in log10 domain (García-García et al., 2004). The averaged seismic moment is $8.80 \times 10^{17}$, leading to a moment magnitude of Mw 5.9, equal to the moment magnitude given by AFAD. The average corner frequency $f_c$ values obtained for P and S-waves are 0.23 Hz and 0.24 Hz, respectively, with a ratio of $\frac{f_{cP}}{f_{cS}} = 0.96$. The obtained ratio of corner frequencies from P- and S- waves is lower than the $\frac{V_P}{V_S} = 1.73$, which holds for a stationary source and can be decreased due to the rupture propagation effects (Sato and Hirasawa, 1973; Kwiatek and Ben-Zion 2013). In general, the $f_{cP} > f_{cS}$ arises for roughly equidimensional source models (L=W). While for long and thin faults, lower $\frac{f_{cP}}{f_{cS}}$ ratios are to be expected; for example, $\frac{f_{cP}}{f_{cS}} = 0.77$ assuming rupture velocity $V_R = 0.9V_S$ (Savage et al., 1972); nearly equal $f_{cP}$ and $f_{cS}$ are given in a dislocation model with a unilateral rupture propagation (Haskell, 1964, Molnar et al., 1973). The small $f_{cP}/f_{cS}$ ratio might imply that the fault width $W$ could be overestimated from the aftershock distribution and could be smaller than 8 km, which is also supported by the narrower fault width estimated from the early aftershock distribution (Fig. 9).

Attenuation greatly affects the amplitude and frequencies included in the seismograms.
Commonly, S waves tend to have larger attenuation than P waves. The ratio between the P and S
quality factors is:

$$\frac{Q_P}{Q_S} = \frac{3}{4}\frac{V_P^2}{V_S^2}. \tag{18}$$

For a Poisson solid, $V_P = \sqrt{3}V_S$, resulting in $\frac{Q_P}{Q_S}$=2.25. Our observations provide a considerably
lower $\frac{Q_P}{Q_S}$=1.2. Such lower ratios are not uncommon and have been interpreted as the attenuating
effect of pore fluids (Olsen et al., 2003; Hauksson and Shearer, 2006; Kwiatek et al., 2013, 2015).
Utilizing the average source size and seismic moment from both P- and S-waves, the static stress
drop of the mainshock is estimated as 0.61 MPa and 1.48 MPa while using a Brune model (Eq 12),
and a Haskell model, respectively (Table 1). The estimated rupture length varies around 14 km
and 6 km for Brune and Haskell model, which yields a rectangular source with a small *L/W* ratio.
A relatively small aspect ratio was also observed for the $M_W$ 1999 7.1 in direct vicinity of this area
(Bürgmann et al., 2002).
Fig. 10 shows P-wave arrivals highlighting the initial portion of the ground displacement record
$\Delta t$. Longer $\Delta t$ rise times and durations of first P-wave displacement pulses are observed for
western stations with azimuth angles of 196°-293°(i.e., stations SUSU, GEYV, KAYN and
KAND). At the same time, eastern stations at comparable distances and azimuth angles ranging
32°-130° display shorter rise times and visibly higher frequency content (see discussion in Douglas
et al., 1988, Fig. 10b), especially for station RUZG and BCAM near 90° azimuth. These
observations suggest eastwards rupture propagation while assuming a unilateral rupture.
However, in the case of a more complicated rupture process, the shorter rise time could also be
promoted by a closer large-local slip asperity in the eastern direction. We also estimated the
azimuthal variations on the $f_c$ for the stations between 200 km and 800 km from the mainshock.
(Fig S9). Larger $f_c$ values are observed at approximately 100°, hence being roughly consistent
with the eastward rupture propagation. However, we note that scattered large $f_c$ values were also
observed at other azimuths.
**Table 1. *Source parameters for the 2022 $M_W$ 6.0 Gölyaka-Düzce earthquake*.** *$f_c$: Corner*
*frequency. $M_0$: Seismic moment. L: Source rupture length. Q: Quality factor. $\Delta\sigma$: Stress drop.*

|  | Average value | Multiplicative error factor |
|---|---|---|
| $M_0$ (N m) | $8.80 \times 10^{17}$ | 2.60 |
| $f_{cP}$ (Hz) | 0.23 | 1.51 |
| $f_{cS}$ (Hz) | 0.24 | 1.52 |
| $Q_P$ | 571 | 1.52 |
| $Q_S$ | 476 | 1.28 |
| $L_{Brune}$ (km) | 14.26 | 1.66 |
| $L_{Haskell}$ (km) | 5.90 | 1.63 |
| $\Delta\sigma_{Brune}$ (MPa) | 0.61 | 2.60 |
| $\Delta\sigma_{Haskell}$ (MPa) | 1.48 | 2.93 |


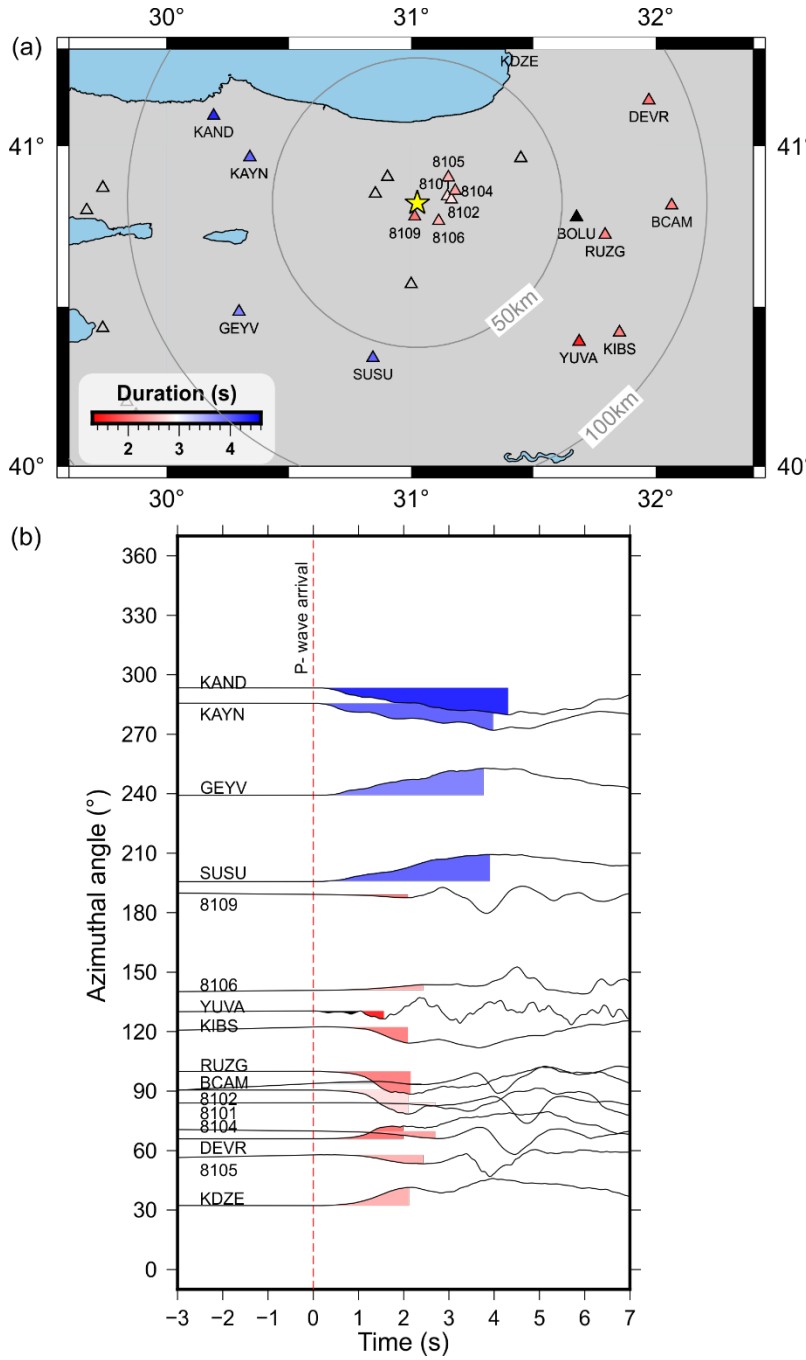

*Figure 10: First-peak duration recorded at seismic stations between 50 km and 100 km from the mainshock epicenter. (a) Station distribution near the epicenter (yellow star). Colored triangles highlight the stations used in (b). (b) Normalized displacement recordings on radial components. The waveforms are aligned relative to the P-wave arrival (0 s) in the time axis and are ordered according to the azimuthal angles relative to the mainshock. The time duration of the colored segments is shown color-coded for in the station symbols in (a).*


## 4 Discussion

The various spatio-temporal scales covered by the different methodologies applied in this study provide insights into the processes leading to and involved in the rupture of the $M_W$ 6.0 Gölyaka-Düzce earthquake. In the following, we discuss the most important patterns that emerged from the

obtained results, as well as their relation with the rheology of the region, the development of previous large earthquakes (i.e. the $M > 7$ Izmit and Düzce earthquakes), and its stage in the seismic cycle.

## 4.1 The 1999 $M > 7$ Izmit and Düzce earthquakes promoted the seismic activation of the Karadere fault

The Karadere fault connects the Akyazi and Düzce basins, which are both pull-apart structures in response to the regional transtensional tectonic setting (Pucci et al., 2006; Ickrath et al., 2015; Bohnhoff et al., 2016b). The spatio-temporal evolution of seismicity along different portions of the broader Marmara region since 1990 shows that the Karadere fault was primarily quiet until the occurrence of the 1999 M>7 Izmit and Düzce events (Fig 5). Most of the Karadere fault was activated during the 1999 August 17th, $M_W$ 7.4 Izmit earthquake while its northeastern portion likely hosted fewer aftershocks (Bohnhoff et al., 2016b). The 1999 Izmit rupture was then extended further eastwards 87 days later with the 1999 November 11th $M_W$ 7.1 Düzce earthquake onto the east-west trending Düzce fault splaying off the Karadere fault, and also dipping towards the North with a dip of around 55° (Bürgmann et al., 2002).

The November 23th 2022 $M_W$ 6.0 Gölyaka-Düzce rupture likely occurred on the northeastern portion of the Karadere fault that remained inactive in 1999, marking the western flank of the Düzce Basin as a topographic depression north of the Düzce fault as a releasing bend. The fact that the Izmit rupture stopped on the Karadere fault and redirected onto the Düzce fault indicates that the northeastern Karadere fault acted as a barrier in 1999. This is supported by the observation of a lower seismic velocity contrast in the Karadere fault with respect to the fault regions west of it (e.g. the Mudurnu fault, see Najdamahdi et al., 2016). Nevertheless, our results show increased background seismic activity from 1999 onwards in the Karadere segment, with a visible increase in 2004-2005. One hypothesis is that the stress redistribution from the 1999 Izmit and Düzce earthquake sequences brought the Karadere segment closer to failure by stress transfer, leading to a progressive activation of this segment over the years. That way, after 23 years of additional continuous tectonic loading, it was finally activated with a $M_W$ 6 event within a region of the fault zone that it still is in a relatively early phase of the seismic cycle. Some support for this scenario comes from a reported change in stress regime together with a rotation of the $S_{hmin}$ orientation in the Karadere segment before and after the 1999 Izmit and Düzce sequences (Ickrath et al., 2015). Before the earthquakes, a predominantly normal faulting stress regime was observed, while strike-slip regime was observed after the Düzce earthquake. As the magnitude of $S_V$ at a certain depth is mostly given by the weight of the overburden, it is expected to remain approximately constant during the earthquake cycle. This suggest that the horizontal shear stresses on the fault increased after the 1999 sequence. We additionally note that the average recurrence period of M > 7 earthquakes in area is around 250 years (Murru et al., 2016). Therefore, the recurrence time of a M > 6 earthquake should be about 25 years, which roughly fits with the occurrence of the last M > 7 earthquakes 23 years before the Gölyaka-Düzce event.

The observed changes in the background seismicity rates could also be related to a change in the seismic coupling of the region (e.g. Marsan et al., 2017; Jara et al., 2017). In particular, the occurrence of the 1999 $M > 7$ Izmit and Düzce earthquakes and their post seismic deformation could have resulted in promoting the occurrence of aseismic slip at depth, hence leading to a progressive decoupling of the fault. The build-up of stresses from the occurrence of enhanced

aseismic slip can increase the background seismicity rates over the region with distributed
deformation over a large area. Indeed, an additional proposed mechanism for the 1999 Düzce
rupture was viscoelastic post-seismic relaxation at depth affecting a broad area from the 1999 Izmit
rupture (e.g. Bürgmann et al., 2002; Ergintav et al., 2009). A detailed study on the microseismicity
from this area also suggested that this possibility could account for the larger seismicity rates at
depths (Beaucé et al, 2022).

## 4.2 How did the mainshock start?

Our catalog of absolute locations revealed at least 23 seismic events with epicentral location less
than 25 km from the $M_W$ 6.0 Gölyaka-Düzce during the month before its occurrence. Out of them,
only two are located in the north eastern segment of the Karadere Fault, as the main fault segment
that ruptured in the Gölyaka-Düzce event. The spatio-temporal evolution of these events does not
suggest clustering, but rather a scattered activation of the area (Fig S7).
Likewise, the foreshocks do not generally resemble a spatial or temporal localization of the
seismicity prior to the mainshock. This is of relevance since a number of moderate to large
earthquakes in this region displayed systematic foreshock activity (Bouchon et al., 2011; Ellsworth
and Bulut, 2018; Malin et al., 2018; Durand et al., 2020). A similarly spatio-temporally scattered
precursory activity pattern as for the mainshock was also found for the 1999 $M_W$ 7.1 Düzce
earthquake, where the largest event in the region of the earthquake rupture in the preceding 65 h
was a $M$ 2.6 event (Wu et al., 2013). Additional small events detected around the future Düzce
1999 rupture did not show any clear signatures of acceleration. The few seismic events preceding
the 2022 $M_W$ 6.0 event, together with their lack of spatio-temporal localization suggest the
existence of relatively homogenous local stress conditions along this fault segment, or
alternatively, homogeneous fault strength, that would allow a progressive fault loading without
rupturing      small heterogeneities in the medium reflecting foreshock activity. In this respect,
laboratory rock deformation experiments have shown that seismic precursors are more frequent
on rough fault surfaces, while seismic foreshocks are much less frequent on polished fault surfaces
(Dresen et al., 2020). This is consistent with the linear and relatively simple geometry of the eastern
portion of the Karadere segment. In fact, the decade-long seismicity along the Karadere fault
shows that it is notoriously more localized within the fault trace than in other fault areas (see e.g.
Wu et al., 2013).
The fault area that was activated in the 1999 M > 7 Izmit and Düzce earthquakes is documented
to continue displaying post-seismic deformation almost 20 years after (Ergintav et al., 2009, Aslan
et al., 2019), mainly related to afterslip and viscoelastic relaxation. In this respect, one possibility
is that the initiation of the mainshock was also promoted by the occurrence of distributed aseismic
slip in the region at depth over a broad area (e.g. Beaucé et al., 2022; Karabulut et al., 2022). This
is supported by the observation of a small number of seismic events around November 11[th], at the
bottom of the Düzce fault, near the place where the 1999 Düzce earthquake nucleated (Fig 6).
Another hypothesis is that a regional or local stress perturbation could have destabilized the
northeastern Karadere fault that was close to failure. Some examples for such a potential stress
perturbation may include tidal effects or seasonal effects such as the effect of precipitation (e.g.
Hainzl et al., 2013) or barometric pressure changes (Martínez-Garzón et al., 2023). Regarding
seasonal or semi-periodic stress perturbations, it is worth to mention that the $M_W$ 6.0 Gölyaka
earthquake, a $M_W$ 4.9 event in 2021 as the largest and most recent event in this area, and the 1999
$M_W$ 7.1 occurred within the second half of November. Further statistical analysis is not conducted
in the frame of this study, but may give further indication on whether earthquakes in this region
show any significant temporal pattern.

### 4.3 Fault segments potentially activated during the mainshock and aftershock sequence

Based on the estimated rupture length from the mainshock source parameters, the event activated a ~12 km long segment of the Karadere fault, terminating just east of the Düzce Basin (Fig 9). Although we tested the application of EGf methods to recover the directivity more accurately, the analysis did not yield clear results (see Text S1 for details). The reasons for this are not clear. It may be that the events used for the EGF deconvolution did not fulfill all necessary criteria (e.g. occurring on the same location, similar focal mechanism and at least a unit of magnitude difference). Alternatively, it could be that the mainshock rupture did not activate a single fault segment, resulting in some complexity obscuring the directivity pattern.

The rupture complexity is also somewhat consistent with the spatial distribution of aftershock seismicity, which shows a heterogeneous event distribution, possibly also illuminating fault structures that were not directly involved in the mainshock rupture. On the western section, the spatial distribution of aftershock seismicity is oriented SE, and part of the distribution suggests the activation of a NW-dipping fault plane of the mainshock in accordance with fault-plane solutions of the event (Table S2), as well as with the size of the mainshock rupture estimated from source parameters.

However, the eastern part of the aftershock distribution is also compatible with the fault geometry of the main Düzce fault activated in 1999. Indeed, the main cluster of events is located at approximately 10 km distance from the mapped surface trace of the Düzce fault. As the deepest aftershock seismicity is located at about 15 km depth, the distribution is also consistent with a fault dipping at about 55°, as we previously reported in Section 3.2.2 (Fig 9). Indeed, this dip is more consistent with the fault geometry reported for the Düzce fault (Bürgmann et al., 2002) than with the dip of the Karadere fault extracted from the focal mechanism of the $M_W$ 6.0 Gölyaka-Düzce earthquake.

Therefore, we suggest that the aftershock distribution that we obtained is likely reflecting the activation of both faults, the Karadere segment displaying a steeper dip towards the northwest, as observed from the focal mechanisms, and the main Düzce fault at depth dipping more gently (around 55°) towards the north.

### 4.4 A proxy for rupture directivity suggests a larger radiation of energy towards the East

Higher frequency P-wave pulses with shorter rise times were identified in the eastward seismic stations from the rupture, suggesting that the mainshock propagated towards the East. A statistically-preferred rupture propagation towards the East was also resolved in the Karadere fault segment below 5 km depth based on the analysis of fault-zone head waves (FZHW) and fault-zone reflected waves (FZRW) (Najdahmadi et al., 2016). At depth, the authors identified the faster side being the elevated crustal Almacik block to the SW. Together with models of bimaterial ruptures, these results suggest that earthquakes on the Karadere segment nucleating at > 5 km depth have a physically explainable preferred propagation direction to the east. However, at shallower depth the fault core was detected to host even slower material between both blocks to either side (Najdahmadi et al., 2016). This led the authors to conclude on a narrow wedge-shaped structure of the fault rather than a simple first-order impedance contrast of the fault. A preferred rupture propagation towards the East was also resolved in the Mudurnu Fault segment (about 70 km West of the mainshock epicenter, see Fig. 6) from detection of fault zone head waves (Bulut et al., 2012). From the moveout of the fault zone head waves, a velocity contrast of about 6% was estimated, with a slower seismic velocity for the northern side of the fault. An eastward propagation of the rupture was also reported for the 1999 $M_W$ 7.1 Düzce earthquake rupture from the joint analysis of geodetic, seismic and strong motion data (Konca et al., 2010). We conclude that based on our

observations of an eastward-directed rupture during the $M$w 6.0 Gölyaka-Düzce earthquake, the observations of the fault-zone head waves in the region and existence of bimaterial faults in the area should be considered an important ingredient for refined seismic risk studies in the area, especially for the Istanbul metropolitan region further to the West. However, this only applies if earthquakes are located along the bimaterial interface. For earthquakes located on secondary splay faults or in the rock volume, their rupture directivities may be related to other factors. A future possible analysis of the source parameters from the smaller events of the sequence may reveal whether the eastward directivity is a persistent feature in the region.

## 5  Conclusions

We investigated the source parameters of the 2022 $M_W$ 6.0 Gölyaka earthquake in NW Türkiye, as well as the evolution of the seismicity framing this mainshock at various spatial and temporal scales. This earthquake mainly ruptured the Karadere fault, a small fault segment located in direct vicinity of the 1999 $M_W$ 7.1 Düzce earthquake. Hence, this case is an example of a medium size earthquake which ruptured a critically stressed fault embedded in a fault zone that is overall in a relatively early stage of the seismic cycle. Our primary goal was to determine how the earthquake initiated, what are the ongoing deformation mechanisms in the region, and how the energy from this mainshock was radiated. The main conclusions extracted from our analysis are the following:

(1) The decade-long evolution of background seismicity in the Karadere segment shows that the segment was mostly silent before the 1999 $M > 7$ Izmit and Düzce earthquakes. From the year 2000 to present, the segment has been comparatively more seismically active, supporting the hypothesis of a progressive approach to critical stress level of the fault segment.

(2) The high-resolution seismicity catalogs derived in this study report on 23 locatable events during the previous month within a 25 km radius from the 2022 Gölyaka-Düzce earthquake. Only few of them occurred close to the future earthquake rupture, suggesting relatively homogenous fault stress conditions and no signatures of foreshock localization were observed.

(3) The early aftershocks of the sequence (i.e. first six days) suggested activation of the Karadere fault segment dipping steeply towards the NW as reported by the moment tensor, and the Düzce fault in the southern part dipping shallower directly towards the North. This suggests that the mainshock rupture, located along the Karadere fault, was able to trigger abundant aftershocks in the neighboring fault segment.

(4) Analysis of mainshock rupture directivity patterns including an attempt to employ empirical Green's functions analysis did not yield clear results. However, shorter rise time and higher frequency content of the P-wave pulses is observed at seismic stations located East of the mainshock hypocenter. If the mainshock rupture did indeed show promoted directivity towards the East, the observation is consistent with predictions from models of bimaterial interfaces and observations from fault zone head waves at this fault.

## Author contribution

PMG and DB generated the enhanced seismicity catalog, JJ analyzed the long-term seismicity rate variations, XC estimated the stress drop, EGF and directivity estimations using methodologies

developed previously by GK, all authors discussed the results in terms of the seismo-tectonic setting, PMG drafted the manuscript and all co-authors contributed to its finalization.

## Acknowledgments

We thank Elif Turker and Fabrice Cotton for discussions and comparison with their directivity results from ground motion. P.M.G. and D.B. acknowledge funding from the Helmholtz Association in the frame of the Young Investigators Group VH-NG-1232 (SAIDAN). X. C. acknowledges funding from the Deutsche Forschungs Gemeinschaft (DFG) in the frame of the proposal "Earthquake source characterization and directivity effects near Istanbul: Implications for seismic hazard".

## Open Research

Seismicity catalogs generated in this study with Artificial intelligence are being prepared for public release through the repository of the GFZ Data Services (link in preparation). While this is being prepared, we provide the three catalogs developed here within the submission files.

Seismicity catalogs from AFAD and KOERI agencies are available under the landing websites https://tdvms.afad.gov.tr/ (last accessed 29/08/2023) and http://www.koeri.boun.edu.tr/sismo/2/earthquake-catalog/ (last accessed 29/08/2023), respectively. The here generated AFAD and KOERI catalogs correspond to the time period from October 23$^{rd}$, 2022 at 00:00h up to November 29$^{th}$, 2022 at 00:00h. Longitude and latitude ranges of 30°-32°, and 40°-41°, respectively.

## Competing interests

The authors declare no conflict of interests.

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

Methods, in Complexity In *Encyclopedia of Complexity and System Science*, Part 5, Springer,
New York, pp. 2449-2473, doi:10.1007/978-0-387-30440-3.
Malin, P. E., Bohnhoff, M., Blümle, F., Dresen, G., Martínez-Garzón, P., Nurlu, M., et al. (2018).
Microearthquakes preceding a M4.2 Earthquake Offshore Istanbul. *Scientific Reports*, *8*(1),
16176. https://doi.org/10.1038/s41598-018-34563-9
Martínez-Garzón, P., Bohnhoff, M., Ben-Zion, Y., & Dresen, G. (2015). Scaling of maximum
observed magnitudes with geometrical and stress properties of strike-slip faults. *Geophysical*
*Research Letters*, 2015GL066478. https://doi.org/10.1002/2015GL066478

Martínez-Garzón, P., Beroza, G. C., Bocchini, G. M., & Bohnhoff, M. (n.d.). Sea level changes
affect seismicity rates in a hydrothermal system near Istanbul. *Geophysical Research Letters*,
*n/a*(n/a), e2022GL101258. https://doi.org/10.1029/2022GL101258
Molnar, P., Tucker, B. E., & Brune, J. N. (1973). Corner frequencies of P and S waves and models
of earthquake sources. *Bulletin of the Seismological Society of America*, *63*(6-1), 2091-2104.
Murru, M., Akinci, A., Falcone, G., Pucci, S., Console, R., & Parsons, T. (2016). M ≥ 7 earthquake
rupture forecast and time-dependent probability for the sea of Marmara region, Turkey. *Journal*
*of Geophysical Research: Solid Earth*, 2015JB012595. https://doi.org/10.1002/2015JB012595

Najdahmadi, B., Bohnhoff, M., & Ben-Zion, Y. (2016). Bimaterial interfaces at the Karadere segment of the North Anatolian Fault, northwestern Turkey. *Journal of Geophysical Research: Solid Earth*, *121*(2), 2015JB012601. https://doi.org/10.1002/2015JB012601

Ogata, Y. (1988). Statistical Models for Earthquake Occurrences and Residual Analysis for Point Processes. *Journal of the American Statistical Association*, *83*(401), 9–27.https://doi.org/10.1080/01621459.1988.10478560

Ogata, Y., & Katsura, K. (1993). Analysis of temporal and spatial heterogeneity of magnitude frequency distribution inferred from earthquake catalogues. *Geophysical Journal International*, *113*(3), 727–738. https://doi.org/10.1111/j.1365-246X.1993.tb04663.x

Özalp, S. and Kürçer, A., 2022. 23 Kasım 2022 Gölyaka (Düzce) Depremi (Mw 6,0) Saha Gözlemleri ve Değerlendirme Raporu, MTA Jeoloji Etütleri Dairesi Başkanlığı, Ankara, Türkiye, 30 s. (*in Turkish*)

Pucci, S., Pantosti, D., Barchi,M., Palyvos, N., 2006. Evolution and complexity of the seismogenic Düzce fault zone (Turkey) depicted by coseismic ruptures, Plio-Quaternary structural pattern and geomorphology. Geophys. Res. Abstr. 8 08339).

Raub, C., Martínez-Garzón, P., Kwiatek, G., Bohnhoff, M., & Dresen, G. (2017). Variations of seismic b-value at different stages of the seismic cycle along the North Anatolian Fault Zone in northwestern Turkey. *Tectonophysics*, *712–713*, 232–248. https://doi.org/10.1016/j.tecto.2017.05.028

Ross, Z. E., Cochran, E. S., Trugman, D. T., & Smith, J. D. (2020). 3D fault architecture controls the dynamism of earthquake swarms. *Science*, *368*(6497), 1357–1361. https://doi.org/10.1126/science.abb0779

Sato, T., and T. Hirasawa (1973), Body wave spectra from propagating shear cracks, *J. Phys. Earth,* 21, 415–432.

Stirling, M. W., Wesnousky, S. G., & Shimazaki, K. (1996). Fault trace complexity, cumulative slip, and the shape of the magnitude-frequency distribution for strike-slip faults: a global survey. *Geophysical Journal International*, *124*(3), 833–868. https://doi.org/10.1111/j.1365-246X.1996.tb05641.x

Olsen, K. B., Day, S. M., & Bradley, C. R. (2003). Estimation of Q for long-period (> 2 sec) waves in the Los Angeles basin. *Bulletin of the Seismological Society of America*, *93*(2), 627-638.

Salvatier, J., Wiecki, T. V., & Fonnesbeck, C. (2016). Probabilistic programming in Python using PyMC3. *PeerJ Computer Science*, *2*(4), e55. https://doi.org/10.7717/peerjcs.55

Sengör, A. M. C. (2005). The North Anatolian Fault: a new look. *Ann. Rev. Earth Panet. Sci.*, *33*, 37–112.

Snoke, J. A. (1987). Stable determination of (Brune) stress drops. *Bulletin of the Seismological Society of America*, *77*(2), 530-538.

Stein, S., & Wysession, M. (2009). *An introduction to seismology, earthquakes, and earth structure*. John Wiley & Sons.

Stierle, E., Vavryčuk, V., Šílený, J., & Bohnhoff, M. (2014). Resolution of non-double-couple components in the seismic moment tensor using regional networks—I: a synthetic case study. *Geophysical Journal International*, *196*(3), 1869–1877. https://doi.org/10.1093/gji/ggt502

Tibi, R., Bock, G., Xia, Y., Baumbach, M., Grosser, H., Milkereit, C., et al. (2001). Rupture processes of the 1999 August 17 Izmit and November 12 Düzce (Turkey) earthquakes. *Geophysical Journal International*, *144*(2), F1–F7. https://doi.org/10.1046/j.1365-246x.2001.00360.x

Utsu, T., & Seki, A. (1954). A relation between the area of after-shock region and the energy of main-shock. *Journal of the Seismological Society of Japan*, *7*, 233–240. https://doi.org/10.4294/zisin1948.7.4233

Waldhauser F., & Ellsworth, W.L. (2000), A double-difference earthquake location algorithm: Method and application to the northern Hayward fault, *Bull. Seism. Soc. Am.*, 90, 1353-1368.

Waldhauser F, Ellsworth W.L., Schaff, D.P., & Cole, A. (2004), Streaks, multiplets, and holes: high-resolution spatiotemporal behavior of Parkfield seismicity. *Geophys. Res. Lett.*,31, no. 18, doi: 10.1029/2004GL020649.

Wesnousky, S. G. (1988). Seismological and structural evolution of strike-slip faults. *Nature*, *335*(6188), 340–343. https://doi.org/10.1038/335340a0

Wiemer, S., Gerstenberger, M., & Hauksson, E. (2002). Properties of the Aftershock Sequence of the 1999 Mw 7.1 Hector Mine Earthquake: Implications for Aftershock Hazard. *Bulletin of the Seismological Society of America*, *92*(4), 1227–1240. https://doi.org/10.1785/0120000914

Wu, C., Meng, X., Peng, Z., & Ben-Zion, Y. (2013). Lack of Spatiotemporal Localization of Foreshocks before the 1999 Mw 7.1 Düzce, Turkey, Earthquake. *Bulletin of the Seismological Society of America*, *104*(1), 560–566. https://doi.org/10.1785/0120130140