# Peer review of "The 2022 MW 6.0 Gölyaka-Düzce earthquake: an example of a medium size earthquake in a fault zone early in its seismic cycle"

_EGUsphere, 2023_

## Author Response (AR1)

Review #1

In this paper, the authors investigated a 2022 M6.0 earthquake sequence along the North Anatolian Fault to understand the preparation, initiation, and the rupture processes of the M6.0 earthquake. They first investigated the existing catalogs with several decades of recorded earthquakes in this area to understand the long-term stress accumulation. They also performed AI-aided earthquake detection, location, and relocation to the recorded waveforms before and after the mainshock to understand the initiation and the mainshock geometry. Moreover, they calculated the earthquake rupture directivity to understand major rupture processes.

The authors performed comprehensive and rigorous analyses of the whole sequence. The content fit the scope of SE and the results are well presented. However, the scientific question is not efficiently demonstrated in the introduction section. Here are more detailed comments.

**Major comments:**

Introduction section: The authors provide a lot of details about the tectonics and historic earthquakes around the study area. However, there is limited description about the importance and the motivation of this work. Why is it necessary to learn this M6.0 earthquake sequence? How does it imply to the occurrence of great earthquakes in this area? How does the seismotectonic setting help us to understand the importance of this sequence? Why are the analyses performed in this study helpful to understand some key scientific questions? It would be better to point out the motivation, main findings, and importance of this study clearly in the introduction section so that the audiences know what to expect in the following sections.

Reply: Thank you for this comment. Indeed, the initial submission did not elaborate much on the primary motivations behind studying this sequence and the key scientific questions addressed. Therefore, in this version we have intensively rewritten the Intro to reflect these topics. To maintain the length of the paper, we removed some paragraphs regarding details on the Gölyaka event and the 1999 Izmit and Düzce sequences that were not totally relevant for the present study. We hope that the new introduction fits better the scopes of the paper and describes the scientific relevance of studying this sequence.

Discussion section: Both the title and the conclusion emphasize that this earthquake sequence is in a fault zone early in its seismic cycle. why it is important? How is it different from other moderate magnitude earthquakes in the other stage of the seismic cycle?

Reply: We aimed at emphasizing that the 2022 M 6 Gölyaka-Düzce event was the largest earthquake in the region after the M > 7 1999 sequence. In this version, we have more explicitly stated about the relatively low elastic strain accumulated on this fault zone in the broad sense, while the segment that ruptured corresponds to a highly stressed segment that did not fail in the previous 1999 sequence. This topic is now more highlighted both in the intro and the discussion sections.

**Minor comments:**

**Introduction section:**

1st Paragraph of Introduction section: Can you provide the full name of "GEOFON, KOERI, USGS, GCMT" somewhere in the main text or supplementary material? It is difficult for audiences to find and check these resources if they do not know these before.

Reply: The corresponding abbreviations and web addresses have been added as footnotes. Note that GEOFON is the full name and not an abbreviation.

[Figure]

3rd paragraph of Introduction section: Where is "Sapanca segment", "Karadere fault", and "Almacik Block"?

Reply: As a reshape of the introduction to better highlight the scientific key questions, these references to places have been removed.

5th paragraph of Introduction section: It would be better to label "Karadere fault" in fig1 so that we can know where the aftershocks are. Is it possible to have a supplementary figure showing the locations of these aftershocks?

Reply: As part of reshaping the intro, this sentence has been moved to Data and Method / Results, where we compare the original (KOERI) and our here derived catalog.

**Data and method section:**

1st paragraph of section 2.1: Can you provide more details about the "decade-long evolution of seismicity"? From which year to which year? How many events in total?

Reply: We thank the reviewer for the comment, and we have updated the analysis timing (1990-2022), including the number of events reported by institutions as well as magnitude ranges.

2nd paragraph in section 2.1: What do you mean by "a better fit to the lower magnitudes of the magnitude-frequency distribution"? How do you define the misfit? Usually, lower $M_c$ may have worse fit when the $M_c$ is lower than the actual $M_c$. Do you mean that this probabilistic approach usually results in a relatively higher $M_c$ so that it fits the curve better? Based on the insets in Fig.2, it seems like this probabilistic approach results in a larger $M_c$ compared with maximum curvature method.

Reply: We acknowledge the point stated by the reviewer. We have changed the manuscript, describing that the method fits the number of earthquakes as a function of magnitudes, employing all event catalogs, removing the statement: "a better fit to the lower magnitudes of the magnitude-frequency distribution." The reviewer rightly mentions that our method is more conservative than the maximum curvature method. Still, our approach has the advantage of using all catalog events to estimate the $M_c$, reason why we think it is more appropriate, even when it provides a higher $M_c$ estimation.

2nd paragraph of section 2.2: Phasenet is trained on labeled waveforms from Northern California instead of southern California.

Reply: Thank you for noticing the mistake ! We have corrected that.

Section 2.2: Could you please provide a comparison between your enhanced catalog and the catalogs from AFAD and KOERI? In this way, we will have a better understanding of the accuracy and the improvements of the new catalog. It seems like Fig 3 shows the comparisons but there is no description in the main text about the details of Fig 3b.

Reply: We have now added some additional sentences describing the initial catalog as follows: "As a comparison, KOERI reported on a total of 505 events 249 aftershocks with magnitudes ≥$M_L$ 0.5 for the same spatio-temporal region analyzed here (Fig 3b). The aftershock epicenters from the KOERI catalog followed the trend of the Karadere fault northeastward of the section activated during the 1999 Izmit rupture. Out of them, 440 correspond to common events from both catalogs."

[Figure]

A closer look to the missing events by our catalog, suggests that most of the events missed by our methodology occurred during the hours following the mainshock. This missing seems to originate both from missing picks with Phasenet, possibly related to low frequency noise associated to the coda of the mainshock as well as from the association of the picks.

[Figure]

**Cover Letter Figure 1**: Comparison of events included in our catalog of detections (Gamma catalog), and those from KOERI catalog. Red circles with black dots represent the events missing in our catalog.

2nd paragraph of Section 2.3: If you use a window length of 25s, you are using both the arrival and the coda of the P- and S-wave. Will it also include surface wave when the epicentral distance is small?

Reply: Both direct arrivals and early coda waves are included in the computed window; given the station we used (200 – 800 km), P- and S-windows generally are not overlapped. For potential surface waves, we checked all S-wave segments and removed those potentially containing surface waves, which shows larger amplitude and would lead to overestimation of seismic moment.

3rd and 4th paragraph of section 2.3: What is "P- or S-wave trains"? In this inversion, do you assume that Q value is constant from the source to the receiver? It seems like the inversion is performed independently at each station. Have you considered uniform M0 at all stations for the same event and uniform Q value for the similar source-receiver path during the inversion? Also, considering that you are trying to fit individual observation to the Brune's model without using EGF here, have you considered the site effect?

Reply: It could be changed to "P- or S-wave". Indeed, we obtain a Q for each observation from each station representing the overall contribution along the entire ray path. Each inversion for 3 independent parameters was performed separately for each station. This selection was made consciously to minimize the potential of bias of source parameters that could be introduced by source-receiver and station-specific issues such as insufficient knowledge on path-related attenuation, site effects or sensor characteristics for a particular station. The outlying results were then removed from the analysis leading to a robust solution. We do not know if the assumption of distance-dependent Q is advocated, and we did not wish to bias the resolved corner frequencies by assuming a specific Q value.

[Figure]

Last paragraph of section 2.3: Are you still use 25s window for analyzing azimuthal variation of ASTF? Usually, the directivity effect is only obvious at the very early stage of P- and S-wave arrivals because multiples and coda waves will smear the directivity effect. If not, how do you choose time window for calculating ASTF?

Reply: Indeed, in the EGF method we tested/investigated shorter time windows (e.g. 10 s and 5 s) to focus on the initial part of the seismogram that should contain source-related processes. We agree that the directivity effect should be obvious at the early portion of P- and S-wave, thus we employ the approach in which we use the initial stage P-wave segments and check the azimuthal variation of them. The inconclusive results from ASTF approach may come from other reasons such as complex source process or limited EGFs. In this version, we now made clear in the Supplementary Text S2 that shorter windows were used.

**Result section:**

Last sentence on Page 10: Figs 5e-5f show that there are less than 10 background events in total in Karadere Fault zone in the past several decades. Therefore, is it possible that the increase of background event rate after 1999 may only be caused by increasing number stations in this region after the 1999 Izmit earthquake? Moreover, there are limited number of background event to evaluate the increase and decrease of background seismicity rate after the 2014 Saros earthquake.

Reply: We appreciate the remark. Unfortunately, we do not have access to the temporal evolution of the network in the area. Even though this information is unavailable, the region is quite close to the one struck by the 1999 Duzce earthquake, which is a well-recorded sequence. In addition, we note that we did not change the magnitude of completeness of the catalog before and after the sequence, and therefore we only worked with event magnitudes that could be detected since 1990. Therefore, we believe in the temporal behavior observed by our results in all segments.

1st paragraph of section 3.3: If we only consider the very early aftershocks (dark dots in Fig. 9b 9c), the width of the fault could be shorter. And the estimated dip angle of the fault plane can also change a lot (larger dip angle along AA' and smaller dip angle along BB')

Reply: Thanks for the suggestion. We add the following sentence in the end of the paragraph: "…The small $fc_P/fc_S$ ratio might imply that the fault width W could be overestimated from the aftershock distribution and could be smaller than 8 km, which is also supported by the narrower fault width estimated from the early aftershock distribution (Fig. 9)"

**Discussion:**

2nd paragraph of section 4.1: Is it possible to calculate the b-value variation in Karadere segment to see whether there is a decrease of b-value with time (possible increase of stress)?

Reply: Due to the limited amount of seismicity on the fault, particularly before 1999, we are not sure that we can reliably estimate the temporal evolution of b-value on the Kadere fault. Also, the magnitude of completeness is very high over the entire time period, and, although there are currently techniques to overcome this problem (i.e. the b+ technique by Van der Elst), we still hesitate about the potential outcome of such analysis with the current seismicity catalog. This can be the subject of a separate analysis at a regional scale with decreased magnitude of completeness.

2nd paragraph of section 4.2: It is difficult to imagine homogeneous local stress in this area considering the fault segments and the fact that this area acts as a barrier (maybe large stress heterogeneity?) during the 1999 Izmit earthquake rupture. Do you have any other evidence indicating homogeneous stress condition? Like consistent focal mechanisms?

Reply: Unfortunately not so many focal mechanisms are available for this region. Örgülü and Aktar (2001) determined some moment tensors for the large aftershocks of the Izmit-Duzce sequence. In the study region, 5 moment tensors are available, 4 consistent with the Duzce fault, and 1 consistent with the Karadere fault. We do not think that this is conclusive towards or against our hypothesis of an homogeneous stress field.

1st paragraph of section 4.3: Another important reason is the ASTF only use less than 4s (fig 10) while source spectra use 25s-time window including a mixture of multiples that suppress the directivity effects.

Reply: Same as the above comment on "Last paragraph of section 2.3".

Last sentence of section 4.4: It seems like the analysis of head waves and the suggestion of bimaterial ruptures may only work along the fault interface with strong material property contrasts. But for earthquakes in the rock volume without significant relative displacement, their rupture directivities cannot be affected by bimaterial interface and may be related to other factors. It is important to note this difference between directivity in the region and along a bimaterial interface.

Reply: Thank you for this important remark, we have added a sentence in the last paragraph of the discussion stating this comment: "However, this only applies if earthquakes are located along the bimaterial interface. For earthquakes located on secondary splay faults or in the rock volume, their rupture directivities may be related to other factors."

**Figure:**

Fig. 1: Y-axis ticks are not uniform with some of them overlapping with each other. Can you also show the scale bar in fig 1b and 1c?

Reply: Thank you , we have corrected the overlapping axes. Instead of directly adding the scale as the figure insets are quite small, we also marked with a rectangle the area represented in (c) to give a better sense of the scale.

Fig 7: Is it better to use the same x ticks for both fig 7a and 7b?

Reply: Good idea, we have adjusted the ticks accordingly.

[Figure]

**Reviewer #2**

In this paper, the authors explore the characteristics of the 2022 Mw 6.0 Gölyaka-Düzce earthquake and its foreshocks and aftershocks by applying several methodologies, including AI-aided techniques, for earthquake detection, location and relocation. The authors also characterize the source parameters of the sequence in terms of corner frequency, stress drop and directivity. According to the author's interpretation, the mainshock occurred on the northeastern termination of the Karadere fault, in a segment that did not rupture during the 1999 events on the North Anatolian fault. By analyzing the seismicity on this segment of the Karadere fault before the occurrence of the 2022 event (from 1990), the authors noticed an increase in background seismicity starting after the 1999 events, which they correlate with the beginning of the tectonic activation of this segment of the Karadere fault. This study is well written and the methods are sound and relevant to this work. In general, observations and results support the author's interpretations. However, I feel that some points of the manuscript should be clarified as suggested in the comments below.

**General                                                                                                          comments:**

Reading the Introduction, and in particular the first part, I am missing the motivations of this work. Like it is written now, it looks to me more like a report about the Gölyaka-Düzce earthquake. I think that the authors should emphasize a little bit more the importance of this work in the Intro. For instance, I find very interesting the fact that this event occurred on a segment that did not rupture in 1999. Even if a particular region has experienced M > 7 events very recently, this does not mean that another potentially damaging event could not occur, mostly if there is a segment of a fault that did not rupture recently. The authors talk about this later in the paper, but I feel that it should be introduced earlier as motivation to study this particular event.

Reply: Indeed, thank you for this comment, which coincides with the main comment from the other Reviewer. In the initial submission we did not elaborate much on the primary motivations behind studying this sequence and the key scientific questions addressed. In this version we have intensively rewritten the Intro to reflect these topics. We also removed some paragraphs regarding details on the Gölyaka event and the 1999 Izmit and Düzce sequences that were not totally relevant for the present study and indeed looked more like a report. The new intro highlights the fact that this earthquake ruptured a portion of the fault that most likely did not rupture in                                the                                1999                                sequence.

One of the main conclusions of this paper is that the Karadere fault experienced an increase in background seismicity starting from 1999, when two M > 7 events occurred in the region. That interpretation comes from the statistical analysis of the seismicity in the Karadere region between 1990 and 2022. While I find the declustering methodology relevant, I struggle with the interpretation of the results. In particular, Figure 5e and 5f show the temporal evolution of the cumulative background seismicity. When I look on the Karadere region, the total amount of events is below five in both catalogs. Of course Figure 5c and Figure 5d show a relative increase, but I believe that the total number of events is too small to consider this increase significant. For example, is there any chance that these events could actually be aftershocks of the 1999 sequence?

Reply: The Reviewer is correct in that the total number of events is small, but we would also like to notice the high magnitude of completeness (Mc 4.1) employed here for the two catalogs from KOERI and AFAD, which reflects the incompleteness during the earlier part of the catalog (e.g. year 1990). As the seismicity represented only contains events above magnitude of completeness, this means that there has been around 5 events with $M_C > 4.1$ in the Karadere segment during the years after 1999, in comparison with none in the previous time. As to whether these events can still be aftershocks from the 1999 earthquakes, this is something really difficult to confirm or reject with total certainty. Nevertheless, if they were aftershocks, we would expect to see their frequency to decrease as time passes, which is not the case with the observed seismicity. Because of these reasons, we believe that this observation is robust. To better clarify this topic, we have added the following

sentences to the Results section: "A change in its seismic behavior is observed afterwards, when this segment experienced an increase of the seismic activity including more than 5 events with M >Mc = 4.1".

I am little bit confused by the title and by all the time the authors refer to a "fault zone early in its seismic cycle". From what I understood the faults early in the seismic cycle should be those that ruptured during the 1999 sequence, and the conclusion of the authors is that the 2022 events did not evolve in a larger event because the surrounding faults already ruptured in 1999 (negative Coulomb stress change). Is that correct? I feel like this point needs some clarification.

Reply: We agree that previously this was a bit awkwardly formulated. In this version, we have more explicitly stated about the relatively low elastic strain accumulated on this fault zone in the broad sense, while the segment that ruptured corresponds to a small highly stressed segment that did not fail in the previous 1999 sequence. This topic is now more highlighted both in the intro and the discussion sections.

**Specific                                                                                                    comments:**

**Introduction**

The authors talk about a Mw 4.9 earthquake that occurred nearby the 2022 Mw 6.0. Could the authors show the location of this event and maybe also the focal mechanism? Also in a supplementary figure if there is no space in the main figures.

Reply: As we were not discussing any other features related to this event, in the process of rewriting the introduction, that sentence has been removed.

There are several names of fault segments and other features (e.g. Sapanca, Karadere, Alamcik Block, Mudurnu branch, etc.) that are in the text but not shown in a map. Please consider showing in a Figure all the places, geological and tectonic features cited in the text.

Reply: In the process of rewriting the Intro, some of these names have disappeared (e.g. Sapanca). For those remaining, we have added them in a refined inset in Figure 1c.

Please include in the text to what acronyms like KOERI or AFAD stands for.

Reply: They are now added as Footnotes 1 and 2.

**Results**

3.1 Could be possible to locate in one of the existing figures the location of the 2014 Mw 6.9 Saros earthquake?

Reply: Unfortunately the 2014 M 6.9 Saros earthquake is outside the area represented in the main map. We have now indicated with a black circle at the inset figure, but there it appears right at the border of the zoomed in area.

The authors write "A change in its seismic behavior is observed afterwards, when this segment experienced a significant increase of the seismic activity." and "Therefore, it is likely that the region was tectonically activated by the earthquake sequence in 1999, and progressively loaded since then, leading to the MW 6.0 Gölyaka-Düzce earthquake 23 years later". Please refer to the General comments about the significance of this increase in seismic activity in the Karadere region.

[Figure]

Reply: In this version, we have explicitly removed the word "significant" as it is commonly associated with the statistical significance. Please see the Reply to the related "General comment" for more information on this topic.

The authors write "The shape of the background rates is different for the AFAD and KOERI catalogs. This difference might be due to the different number of seismic stations operated by the agencies in this area, hence affecting the monitoring capabilities and detection threshold." Looking at Figure 5, in general the KOERI catalog seems to show a higher background rate.

Reply: Yes. This is because, generally, the seismic network by KOERI used to be larger than that by AFAD until recently. However, we would prefer to keep this statement rather open, as for selected regions and time periods, the coverage by AFAD network might also be better.

3.2 The authors write "Calculating the b-value for events above MC, we find a value of $b$ = 0.95 ± 0.05 (Fig. S6). This b-value is comparatively lower than those obtained for other aftershock sequences (e.g. Wiemer et al., 2002).". Probably is not going to change your results but please clarify that your b-values includes also the foreshocks. Also, this statement seems to be only relative to Wiemer et al. (2002) and the 1999 Hector Mine earthquake. However, the b-value of aftershocks sequence seem to be highly variable (see van der Elst et al, 2022).

Reply: Thank you for the correction. Indeed, the b-value from aftershock sequences can be much more variable than what suggested by that sentence. We have now corrected it by stating: "*Calculating the b-value for events above $M^C$, from both the periods before and after the mainshock, we find a value of $b = 0.95 \pm 0.05$ (Fig. S6).*"

The authors write "The southwestern segment of the Karadere fault was activated in the Izmit event and hosted numerous aftershocks from both the 1999 Izmit and Düzce earthquakes, while the northeastern part of the Karadere fault (now activated in the Gölyaka-Düzce event) hosted fewer aftershocks in 1999 (see seismicity from Bohnhoff et al., 2016b plotted in Fig. 6)." It is a bit hard to see in Figure 6 that the northeastern part of the Karadere fault hosted fewer aftershocks from the 1999 earthquakes. The northeastern segment of the Karadere fault terminates around longitude 31.1°, and the seismicity seems to decrease after that point.

Reply: It is true that this point was not so clearly shown from Figure 6. Hence, we have decided to remove the corresponding sentence.

**Discussion**

4.2 The authors write "The few seismic events preceding the 2022 MW 6.0 event, together with their lack of spatio-temporal localization suggest the existence of relatively homogenous local stress conditions along this fault segment, that would allow a progressive fault loading without rupturing many small heterogeneities in the medium reflecting foreshock activity." This is an interesting point. Could the authors provide some references which relate spatiotemporal distribution of foreshocks and stress distribution?

Reply: We have now broadened the sentence to reflect that indeed the homogenous conditions might relate to stress or to strength. With respect to the Reviewer's comment, we have added the following sentence: "In this respect, laboratory rock deformation experiments have shown that seismic precursors are more frequent on rough fault surfaces, while seismic foreshocks are much less frequent on polished fault surfaces (Dresen et al., 2020)."

[Figure]

4.3 The authors write "Therefore, we suggest that the aftershock distribution that we obtained is likely reflecting the activation of both faults, the Karadere segment displaying a steeper dip towards the northwest, as observed from the focal mechanisms, and the main Düzce fault at depth dipping more gently (around 55°) towards the north." Both focal mechanism and earthquake distribution seems to show that the northeastern segment of the Karadere fault dips to the northwest with has mainly a strike-slip kinematic with a small normal component. However, looking at the topography of the area I would mostly expect a reverse component rather than normal (northwest block up). What is the authors' opinion? Could this just be related to the uncertainties of the focal mechanism solution?

Also, I will be curious to see the 2022 aftershocks on the Düzce fault are particularly located in a part of the fault that experienced small cosiemsic slip or small aftershock activity during the 1999 sequence.

Reply: Yes, reviewer is correct, from the topography the Karadere fault could be more consistent with strike-slip with a small reverse component. However, this does not appear to be the case with the mainshock focal mechanism. We did some research about earthquakes in the area with moment tensor available, and unfortunately not so many focal mechanisms are available. Örgülü and Aktar (2001) determined some moment tensors for the large aftershocks of the Izmit-Duzce sequence. In the study region, 5 moment tensors are available, 4 consistent with the Duzce fault, and 1 consistent with the Karadere fault. All of them displayed strike-slip kinematics. With respect to the focal mechanism uncertainty, it is really difficult to evaluate. Both the mainshock solution as well as those from Örgulu and Aktar (2001) were obtained using full waveform inversion, and hence they should not be as affected by lack of nearby stations as inversion from first motion polarities only.

About the aftershocks on the Düzce fault, it is a good point to compare their location with the slip distribution of the 1999 earthquake. However, after comparing with the results from Konca et al., (2010) it is not so clear. The aftershock area of the Düzce fault is relatively small, and from it, it appears that the aftershocks could be occurring at the boundary of the Düzce rupture, but it is not so easy to accurately compare them.

Figure 1. This figure has some problems with the y-axis of the coordinates.

Reply: We have corrected that, thank you.

Figure 3. I have the feeling that this figure, and in particular because of the choice of colors, maybe difficult to see for people affected by color blindness

Reply: We have now changed the color code of the picture to grayscale, to make it more suitable to color-blind people.

Figure 5. Would it be possible to use the same range of values for the y-axis between the AFAD and KOERI catalogs? I think it would significantly help the reader visualize the difference between the two catalogs.

Reply: We are not so sure about this suggestion. We think that we are not necessarily interested in highlighting the absolute number of events, but rather to highlight the different behavior of individual fault segments before and after large earthquakes, and specifically the 1999 Izmit and Duzce. For that, adjusting the y scale to highlight each particular catalog provides a better visualization.

References

van der Elst, N. J. (2021). B-positive: A robust estimator of aftershock magnitude distribution in transiently incomplete catalogs. Journal of Geophysical Research: Solid Earth, 126, e2020JB021027. https://doi.org/10.1029/2020JB021027